# Projection-Manifold Regularized Latent Diffusion for Robust General Image Fusion

**Lei Cao[1], Hao Zhang[1,2,*], Chunyu Li[1], Jiayi Ma[1,*]**
[1]Electronic Information School, Wuhan University, Wuhan, China
[2]Suzhou Institute, Wuhan University, Suzhou, China
whu.caolei@whu.edu.cn, {zhpersonalbox, licy0089, jyma2010}@gmail.com

## Abstract

This study proposes PDFuse, a robust, general training-free image fusion framework built on pre-trained latent diffusion models with projection–manifold regularization. By redefining fusion as a diffusion inference process constrained by multiple source images, PDFuse can adapt to varied image modalities and produce high-fidelity outputs utilizing the diffusion prior. To ensure both source consistency and full utilization of generative priors, we develop novel projection–manifold regularization, which consists of two core mechanisms. On the one hand, the Multi-source Information Consistency Projection (MICP) establishes a projection system between diffusion latent representations and source images, solved efficiently via conjugate gradients to inject multi-source information into the inference. On the other hand, the Latent Manifold-preservation Guidance (LMG) aligns the latent distribution of diffusion variables with that of the sources, guiding generation to respect the model's manifold prior. By alternating these mechanisms, PDFuse strikes an optimal balance between fidelity and generative quality, achieving superior fusion performance across diverse tasks. Moreover, PDFuse constructs a canonical interference operator set. It synergistically incorporates it into the aforementioned dual mechanisms, effectively leveraging generative priors to address various degradation issues during the fusion process without requiring clean data for supervising training. Extensive experimental evidence substantiates that PDFuse achieves highly competitive performance across diverse image fusion tasks. The code is publicly available at `https://github.com/Leiii-Cao/PDFuse`.

## 1   Introduction

Due to hardware restrictions and optical imaging constraints, single-sensor acquisition systems often fail to capture scene details comprehensively. Consequently, image fusion technology has emerged to meet the demands of various application scenarios [55]. Typical image fusion techniques include infrared-visible image fusion (IVF) [18, 30], multi-exposure image fusion (MEF) [47, 21] and multi-focus image fusion (MFF) [28]. These techniques demonstrate widespread applications in intelligent driving, medical diagnostics, and computational photography [19, 53].

In recent years, the application of advanced feature representations and model architectures to image fusion has enabled task-specific methods to achieve encouraging results [31, 58, 22]. However, approaches that generalize across multiple tasks [48, 67, 49] and maintain robustness to degradation [52, 56, 59] remain relatively scarce. We summarize two primary challenges. **(1) Generality challenge:** advanced universal image fusion techniques [48, 67] primarily depend on custom adaptation networks or memory modules to facilitate cross-task generalization through continual learning.

---

* Corresponding authors.

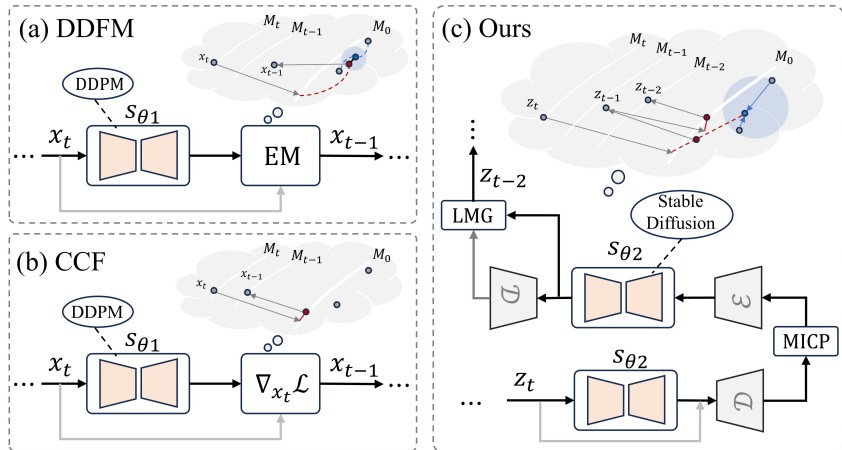

Figure 1: Framework diagrams of current training-free image fusion methods based on pre-trained diffusion models, where (a) illustrates the DDFM [65] framework, (b) illustrates the CCF [3] framework, and (c) illustrates our proposed PDFuse framework.

However, they require retraining for each task scenario, and in fact, their continual learning networks and memory modules merely average knowledge across scenes, so universality remains insufficient. **(2) Robustness challenge:** current approaches build supervision [56] by creating training pairs of clean and degraded images through degradation removal and addition, but their robustness depends heavily on data availability and typically targets only specific degradation types [57]. Therefore, developing a framework that is both robust and truly universal for image fusion remains worth investigating.

By combining diffusion posterior sampling with pretrained diffusion priors [37, 16], pretrained latent diffusion models [12, 36] have demonstrated strong potential across a wide range of domains [44, 68], offering a new perspective for tackling the generality and robustness challenges in image fusion. However, fusion methods based on pretrained diffusion models remain in their infancy. DDFM [65] employs an EM-based likelihood constraint for multimodal fusion, as shown in Fig. 1(a), while CCF[3] uses a conditional library of fusion metrics to guide the diffusion process, as shown in Fig. 1(b). Both focus on applying DDPM[12] to fusion tasks; however, comprehensive investigations of latent diffusion models with enhanced generative capacities and richer priors are lacking, and neither effectively coordinates multi-source information injection with the utilization of diffusion generative priors to overcome the two major challenges mentioned above.

To overcome the limitations of current approaches, we introduce PDFuse, a training-free, universal, and robust image fusion framework that harnesses the generative priors of a pre-trained latent diffusion model, as shown in Fig. 1(c). To our knowledge, PDFuse is the first training-free image fusion approach built on a pre-trained latent diffusion model. During the generative diffusion posterior sampling process, we introduce two key regularization mechanisms to efficiently integrate multi-source image information and leverage generative priors. First, the Multi-source Information Consistency Projection (MICP) establishes a projection system between diffusion latents and source images, solved efficiently via conjugate gradients to inject multi-source information and ensure high fidelity. Second, the Latent Manifold-preservation Guidance (LMG) aligns the latent distribution of diffusion variables with that of the sources, guiding generation to respect the model's manifold prior and further improve fusion quality. By alternating these two mechanisms, PDFuse strikes an optimal balance between fidelity and generative quality. Moreover, PDFuse constructs a canonical interference operator set and integrates it into the dual mechanisms to handle degradations via generative priors without clean-data supervision, significantly enhancing robustness under varied degradation scenarios.

The primary contributions of this study are outlined as follows:

• This study proposes PDFuse, the first training-free, universal, and robust image fusion framework built on a pre-trained latent diffusion model, redefining fusion as a diffusion inference process constrained by multiple source images.

• We design two regularization mechanisms, the Multi-source Information Consistency Projection and the Latent Manifold-preservation Guidance, which are alternately optimized to constrain the diffusion process, enabling PDFuse to adapt to diverse image modalities and generate high-fidelity outputs using the diffusion prior.
• We construct a canonical interference operator set and embed it into both regularization mechanisms, enabling the framework to handle various degradation issues during fusion without clean-image supervision.
• We conducted extensive and comprehensive evaluations of our method. Fusion experiments across multiple task scenarios, along with robustness tests under interference, semantic experiments, and ablation studies, demonstrate the high competitiveness of our approach.

## 2 Related work

**Image fusion.** Since the advent of deep learning [55, 64], fusion frameworks with learnable parameters such as autoencoders [18], convolutional neural networks [63], generative adversarial networks [30, 47], Transformers [31, 58] and diffusion models [54, 51, 41] have dominated image fusion. U2Fusion [48] introduced the first universal fusion framework based on information metrics and continual learning, while TC-MoA [67] employs fine tuned adapters for multi task adaptation but remains limited by the performance of adapters and continual learning networks. Text-IF [52] integrates text guidance to address specific scene degradations and Text-DiFuse [56] couples diffusion and fusion networks to improve robustness to degradation; both methods depend on simulation based degradation data. Although DDFM [65] employs conditional likelihood to leverage pre-trained diffusion models for image fusion, and CCF [3] proposes a flexible, controllable condition set, both remain confined to pre-trained DDPMs, lack exploration of latent diffusion models, and cannot synergistically integrate multi-source conditional embeddings with generative prior exploitation.

**Zero-shot applications of latent diffusion models.** The generative capacities and generative diffusion priors of latent diffusion models open new avenues for zero-shot applications in diverse downstream tasks [68]. Some methods harness diffusion denoising network attention maps for tasks such as image editing [16]. Meanwhile, adaptive posterior sampling strategies have shown significant promise in areas such as image restoration and protein generation [37, 11]. Since image fusion along this trajectory remains in its infancy, exploring training-free applications of latent diffusion models in image fusion and fully exploiting their generative diffusion priors is crucial.

## 3 Preliminaries

Latent Diffusion models transform data samples $\boldsymbol{z}_0 \sim p_{\text{data}}(\boldsymbol{z}_0)$ through a forward Itô stochastic differential equation (SDE) [39]:

$$\mathrm{d}\boldsymbol{z}_t = -\frac{1}{2}\beta(t)\boldsymbol{z}_t \, \mathrm{d}t + \sqrt{\beta(t)} \, \mathrm{d}\boldsymbol{w}_t, \qquad \boldsymbol{z}_0 = \boldsymbol{z}_{t=0}, \tag{1}$$

where $\boldsymbol{z}_t \in \mathbb{R}^k$ represents the $k$-dimensional latent state at time $t$, $\boldsymbol{w}_t$ is a standard $k$-dimensional Brownian motion, $\beta : [0, T] \to \mathbb{R}^+$ defines the noise schedule, and $\mathbf{I}_k \in \mathbb{R}^{k \times k}$ denotes the $k$-dimensional identity matrix. For any $0 \le s < t \le T$, the conditional distribution satisfies [12]:

$$\boldsymbol{z}_t \mid \boldsymbol{z}_s \sim \mathcal{N}\left(\mu(t, s)\boldsymbol{z}_s, \Sigma(t, s)\mathbf{I}_k\right), \tag{2}$$

with signal attenuation $\mu(t, s) = \exp\left(-\frac{1}{2}\int_s^t \beta(u) \, \mathrm{d}u\right)$ and noise accumulation $\Sigma(t, s) = \int_s^t \beta(u) e^{-2\left(\int_u^t \beta(v) \, \mathrm{d}v\right)} \mathrm{d}u$, demonstrating exponential decay characteristics. The corresponding reverse-time SDE, which denoises $\boldsymbol{z}_T \sim \mathcal{N}(\mathbf{0}, \mathbf{I}_k)$ to $\boldsymbol{z}_0 \sim p_{\text{data}}$, obeys:

$$\mathrm{d}\boldsymbol{z}_t = \left[-\frac{1}{2}\beta(t)\boldsymbol{z}_t - \beta(t)\nabla_{\boldsymbol{z}_t} \log p_t(\boldsymbol{z}_t)\right] \mathrm{d}t + \sqrt{\beta(t)}\mathrm{d}\bar{\boldsymbol{w}}_t, \tag{3}$$

where $\bar{\boldsymbol{w}}_t$ denotes reverse-time Brownian motion. A neural network $s_\theta(\boldsymbol{z}_t, t)$ approximates the score function $\nabla_{\boldsymbol{z}_t} \log p_t(\boldsymbol{z}_t)$ via denoising score matching [39]. To sample from the posterior distribution $p(\boldsymbol{z}_0|\boldsymbol{y}) \propto p(\boldsymbol{y}|\boldsymbol{z}_0)p_{\text{data}}(\boldsymbol{z}_0)$ under observation $\boldsymbol{y}$, we modify the drift term of the reverse-time SDE

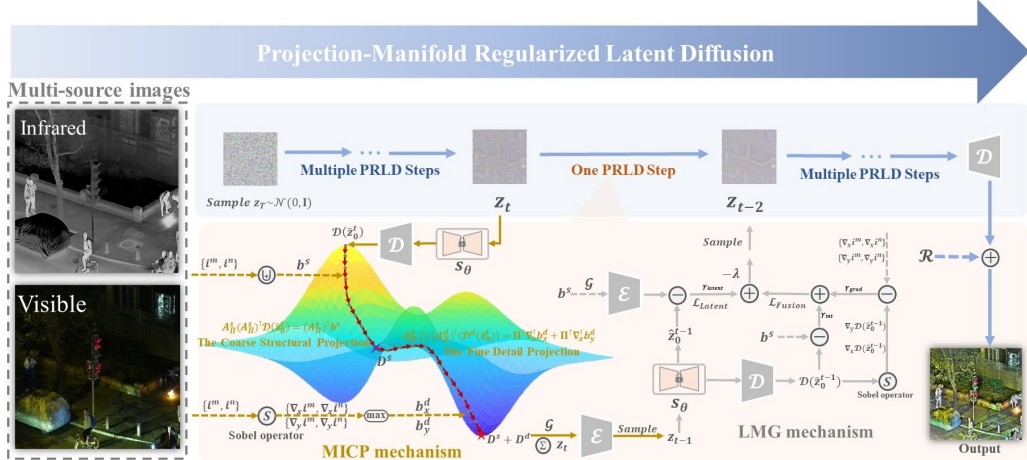

Figure 2: A diagram depicting the inference architecture of Projection–Manifold Regularized Latent Diffusion shows that each PRLD step comprises an alternating optimization cycle between the MICP mechanism and the LMG mechanism.

to incorporate observational constraints [4]:

$$\mathrm{d}\boldsymbol{z}_t = \left[ -\frac{1}{2}\beta(t)\boldsymbol{z}_t - \beta(t)\Big(s_\theta(\boldsymbol{z}_t, t) + \nabla_{\boldsymbol{z}_t}\log p(\boldsymbol{y}|\boldsymbol{z}_t)\Big) \right]\mathrm{d}t + \sqrt{\beta(t)}\mathrm{d}\bar{\boldsymbol{w}}_t, \tag{4}$$

where $\nabla_{\boldsymbol{z}_t}\log p(\boldsymbol{y}|\boldsymbol{z}_t)$ derives from $-\nabla_{\boldsymbol{z}_t}\mathcal{L}$ under the exponential family assumption $p(\boldsymbol{y}|\boldsymbol{z}_t) \propto e^{-\mathcal{L}}$. The discretized update rule via Euler-Maruyama [17] scheme becomes:

$$\boldsymbol{z}_{t-\Delta t} = \boldsymbol{z}_t + \Delta t\left[ -\frac{1}{2}\beta(t)\boldsymbol{z}_t - \beta(t)\left(s_\theta(\boldsymbol{z}_t, t) - \nabla_{\boldsymbol{z}_t}\mathcal{L}\right) \right] + \sqrt{\beta(t)\Delta t}\,\boldsymbol{\epsilon}, \quad \boldsymbol{\epsilon} \sim \mathcal{N}(\mathbf{0}, \mathbf{I}). \tag{5}$$

By this approach, no additional training of the diffusion model is necessary to achieve conditional diffusion via posterior sampling.

## 4    Methodology

Denote the multi-source image inputs as $i^m \in \mathbb{R}^{H \times W \times C}$ and $i^n \in \mathbb{R}^{H \times W \times C}$, and assume a pre-trained latent diffusion model consisting of an encoder $\mathcal{E}(\cdot)$, a decoder $\mathcal{D}(\cdot)$, and a latent noise–prediction network $s_\theta(\cdot)$. Following Eq. (4), we reformulate image fusion as a diffusion inference process constrained by multiple source images. At the t-th sampling iteration, two critical mechanisms are integrated: the multi-source information consistency projection mechanism $\mathcal{F}_\theta(\cdot)$ and the latent manifold-preservation guidance mechanism $\mathcal{P}_{\mathcal{M}}(\cdot)$, defined as:

$$\boldsymbol{z}_{t-1} = \sqrt{\bar{\alpha}_{t-1}}\,\underbrace{\mathcal{F}_\theta\big(\hat{\boldsymbol{z}}_0^t, \{\boldsymbol{y}_i\}_{i=1}^2\big)}_{\text{(MICP mechanism)}} + \sqrt{1 - \bar{\alpha}_{t-1} - \sigma_t^2}\,s_\theta(\boldsymbol{z}_t, t) - \lambda\underbrace{\mathcal{P}_{\mathcal{M}}\big(\hat{\boldsymbol{z}}_0^t, \{\boldsymbol{y}_i\}_{i=1}^2\big)}_{\text{(LMG mechanism)}} + \sigma_t\boldsymbol{\epsilon}. \tag{6}$$

Here, the cumulative noise attenuation coefficient is defined as $\bar{\alpha}_t = \prod_{s=1}^t (1 - \beta_s)$, and the noise modulation term is specified by $\sigma_t = \eta\sqrt{\frac{1-\bar{\alpha}_{t-1}}{1-\bar{\alpha}_t}\left(1 - \frac{\bar{\alpha}_t}{\bar{\alpha}_{t-1}}\right)}$ where $\eta \in [0,1]$. For the multi-source image set $\{\boldsymbol{y}_i\}_{i=1}^2 = \{i^m, i^n\}$, the parameter $\lambda$ governs the strength of the manifold constraint. The clean signal estimate at iteration $t$, denoted $\hat{\boldsymbol{z}}_0^t$, is obtained via Tweedie's formula [12] $\hat{\boldsymbol{z}}_0^t = \frac{\boldsymbol{z}_t - \sqrt{1-\bar{\alpha}_t}\,s_\theta(\boldsymbol{z}_t,t)}{\sqrt{\bar{\alpha}_t}}$. In the following, we provide a detailed exposition.

### 4.1    Alternating optimization with projection strategy and manifold constraints

**Multi-source information consistency projection mechanism.**    As formulated in Eq. (6), the proposed MICP mechanism embeds multi-source image information via the efficient projection of $\hat{\boldsymbol{z}}_0^t$

onto the constrained manifold spanned by the multi-source image set $\{\boldsymbol{y}_i\}_{i=1}^2 = \{i^m, i^n\}$. Inspired by [31], the fusion constraint loss is concisely expressed as follows:

$$\mathcal{L}_{\text{Fusion}}\big(\mathcal{D}(\hat{\boldsymbol{z}}_0^t)\big) = \gamma_{\text{int}}\|\,\mathcal{D}(\hat{\boldsymbol{z}}_0^t) - \max(i^m, i^n)\,\|_2^2 + \gamma_{\text{grad}}\|\,\nabla\mathcal{D}(\hat{\boldsymbol{z}}_0^t) - \max(\nabla i^m, \nabla i^n)\,\|_2^2, \quad (7)$$

where $\max(\cdot)$ denotes the element-wise maximum, $\nabla$ represents the gradient operator, and $\gamma_{\text{int}}, \gamma_{\text{grad}}$ are coefficients balancing the intensity and gradient loss terms. To improve optimization efficiency, we introduce a two-stage projection strategy, as illustrated in Fig. 2. This strategy decomposes the projection into a coarse structural component $\mathcal{D}^s(\hat{\boldsymbol{z}}_0^t)$, which captures the overarching shape and form, and a fine detail component $\mathcal{D}^d(\hat{\boldsymbol{z}}_0^t)$, which refines texture and local details, hereafter denoted as $\mathcal{D}^s$ and $\mathcal{D}^d$, respectively.

In the first stage, we solve the unconstrained quadratic program $\min_{\mathcal{D}(\hat{\boldsymbol{z}}_0^t)} \|\mathcal{D}(\hat{\boldsymbol{z}}_0^t) - \max(i^m, i^n)\|_2^2$, which is designed to ensure a coarse aggregation of the pixel-wise structures from multiple sources. By computing and analyzing the partial derivatives of this objective function, the resulting optimality conditions can be written compactly as the following projection system:

$$(\boldsymbol{A}^s)^\top \boldsymbol{A}^s \, \mathcal{D}(\hat{\boldsymbol{z}}_0^t) = (\boldsymbol{A}^s)^\top \boldsymbol{b}^s, \quad \text{where } \boldsymbol{A}^s = \mathbf{I}. \tag{8}$$

Here, $\mathbf{I}$ denotes the identity matrix, and $\boldsymbol{b}^s = \{i^m \uplus i^n\} \in \mathbb{R}^{H \times W \times C}$. For instance, in infrared-visible image fusion, $\uplus$ represents the $\max(\cdot)$ operator. By employing the conjugate gradient method, we efficiently obtain the coarse structural component $\mathcal{D}^s$, which is produced by the first-stage projection of $\mathcal{D}(\hat{\boldsymbol{z}}_0^t)$.

In contrast to conventional fusion strategies that jointly balance structure and detail, our method explicitly separates texture enhancement as an additive refinement layer. During the second stage, building upon the retained coarse structural component $\mathcal{D}^s$, we optimize and add a fine detail component $\mathcal{D}^d$ to recover high-frequency texture information from the multi-source images $\{\mathbf{y}_i\}_{i=1}^2$. The detail projection follows two key principles: **(i)** maximize the preservation of source image textures, and **(ii)** minimize interference with the previously embedded structural information. Accordingly, by extending the texture loss in Eq. (7), the optimization objective for $\mathcal{D}^d$ can be expressed as:

$$\min_{\mathcal{D}^d}\left\{\underbrace{\left\|\nabla(\mathcal{D}^s + \mathcal{D}^d) - \max(\nabla i^m, \nabla i^n)\right\|_2^2}_{\text{Detail-constraint term}} + \underbrace{\omega\,\|\mathcal{D}^d\|_2^2}_{\text{Regularization term 1}} + \underbrace{\phi\,\|\nabla^2\mathcal{D}^d\|_2^2}_{\text{Regularization term 2}}\right\}. \tag{9}$$

Here, the detail-constraint term enforces texture consistency with the multi-source images and thus implements principle **(i)**, whereas the first regularizer penalizes excessive modifications to the coarse structural component and thereby enforces principle **(ii)**. The second regularizer suppresses rasterization artifacts by penalizing the Laplacian of the detail component. The nonnegative weights $\omega$ and $\phi$ trade off these objectives to maximize texture fidelity subject to structural preservation. By differentiating Eq. (9) with respect to $\mathcal{D}^d$ and rearranging the resulting stationarity condition, we obtain the following projection system:

$$\boldsymbol{A}^d \odot (\boldsymbol{A}^d)^\top \mathcal{D}^d = \nabla_x^\top \boldsymbol{b}_x^d + \nabla_y^\top \boldsymbol{b}_y^d,$$

$$\text{where } \boldsymbol{A}^d \odot (\boldsymbol{A}^d)^\top = \left(\omega\mathbf{I} + \nabla_x^\top\nabla_x + \nabla_y^\top\nabla_y + \phi(\nabla^2)^\top\nabla^2\right), \tag{10}$$

$$\text{where } \boldsymbol{b}_x^d = \max\big(\nabla_x i^m, \nabla_x i^n\big) - \nabla_x\mathcal{D}^s, \quad \boldsymbol{b}_y^d = \max\big(\nabla_y i^m, \nabla_y i^n\big) - \nabla_y\mathcal{D}^s,$$

where $\nabla_x$ and $\nabla_y$ denote the horizontal and vertical gradient operators, respectively. This projection system can be efficiently solved using the conjugate gradient method.

Owing to the VAE's inherent insensitivity to high-frequency components, we apply a Gaussian filter $\mathcal{G}(\cdot)$ to the outputs of the projection system as a preprocessing step. The high-frequency residual $\mathcal{R}$ obtained from this filtering is then added back to the final generated result to restore fine-grained details. Accordingly, the multi-source information consistency projection mechanism can be formulated as follows:

$$\mathcal{F}_\theta\big(\hat{\boldsymbol{z}}_0^t, \{\boldsymbol{y}_i\}_{i=1}^2\big) = (\mathcal{E} \circ \mathcal{G})\Big((1 - \bar{\alpha}_t)(\mathcal{D}^s + \mathcal{D}^d) + \bar{\alpha}_t\,\hat{\boldsymbol{z}}_0^t\Big). \tag{11}$$

**Latent manifold-preservation guidance mechanism.** The LMG mechanism is a posterior sampling process based on a latent diffusion model. By applying the VAE encoder and decoder to both

the multi-source images $\{\boldsymbol{y}_i\}_{i=1}^2$ and $\hat{\boldsymbol{z}}_0^t$, we compute residuals in both the image domain and the latent domain. We backpropagate the corresponding gradients and combine them to update and refine $\hat{\boldsymbol{z}}_0^t$. Consequently, the LMG operator can be written as:

$$\mathcal{P}_{\mathcal{M}}\big(\hat{\boldsymbol{z}}_0^t, \{\boldsymbol{y}_i\}_{i=1}^2\big) = \nabla_{\boldsymbol{z}_t}\mathcal{L}_{\text{Fusion}}\big(\mathcal{D}(\hat{\boldsymbol{z}}_0^t)\big) + \gamma_{\text{latent}}\,\nabla_{\boldsymbol{z}_t}\big\|\hat{\boldsymbol{z}}_0^t - \mathcal{E}(\max(i^m, i^n))\big\|_2^2. \quad (12)$$

$\mathcal{L}_{\text{Fusion}}$ is a pixel-domain loss that aligns $\hat{\boldsymbol{z}}_0^t$ with the multi-source images $\{\boldsymbol{y}_i\}_{i=1}^2$. This loss comprises both intensity and texture terms. A complementary term, inspired by [37], serves as a latent-space regularizer designed to suppress biases introduced by pixel-space guidance. The scalar coefficient $\gamma_{\text{latent}}$ controls the strength of this latent regularization.

**Alternating optimization.** Although the LMG mechanism can guide the latent diffusion process via posterior sampling to embed information from the multi-source image set $\{\boldsymbol{y}_i\}_{i=1}^2$, this guidance often incurs significant information loss and degrades the fidelity of the generated images, as discussed in our experiments. Therefore, to balance effective information embedding with latent manifold-preservation guidance, we adopt an alternating-optimization scheme that switches between the MICP mechanism and the LMG mechanism. The optimization strategy is as follows:

$$(\mathcal{F}_\theta(\cdot), \mathcal{P}_{\mathcal{M}}(\cdot)) = \begin{cases} \big(\mathcal{F}_\theta\big(\hat{\boldsymbol{z}}_0^t, \{\boldsymbol{y}_i\}_{i=1}^2\big),\, 0\big), & \text{if (MICP step)}, \\ \big(\hat{\boldsymbol{z}}_0^t,\, \mathcal{P}_{\mathcal{M}}\big(\hat{\boldsymbol{z}}_0^t, \{\boldsymbol{y}_i\}_{i=1}^2\big)\big), & \text{if (LMG step)}. \end{cases} \quad (13)$$

### 4.2 Integrated interference operator set

We assemble a canonical interference operator set of $N$ typical operators, $\{\boldsymbol{\Pi}_i\}_{i=1}^N$, including overexposure and underexposure, Gaussian blur, motion blur, bicubic downsampling, *etc*. By integrating the set of degradation operators through both mechanisms, our framework alleviates degradation artifacts in image fusion without requiring paired data to train a degradation-inversion model. When the degradation types $\boldsymbol{\Pi}_i$ and $\boldsymbol{\Pi}_j$ are respectively applied to the input images $\{\boldsymbol{y}_i\}_{i=1}^2$, referring to Eq. (8), the first-stage projection strategy of the MICP mechanism, after integrating the set of degradation operators, can be expressed as follows:

$$(\boldsymbol{A}_{\boldsymbol{\Pi}}^s)^\top \boldsymbol{A}_{\boldsymbol{\Pi}}^s \mathcal{D}(\hat{\boldsymbol{z}}_0^t) = (\boldsymbol{A}_{\boldsymbol{\Pi}}^s)^\top \boldsymbol{b}^s. \quad (14)$$

Here, $\boldsymbol{\Pi}$ denotes the aggregation of $\boldsymbol{\Pi}_i$ and $\boldsymbol{\Pi}_j$, and its combination with the corresponding images is related to the fusion weight coefficients at the pixel and texture levels. Building upon Eq. (10), an upon incorporating the degradation-operator set, the MICP mechanism's second-stage projection strategy can be succinctly formulated as:

$$\boldsymbol{A}_{\boldsymbol{\Pi}}^d \odot (\boldsymbol{A}_{\boldsymbol{\Pi}}^d)^\top \mathcal{D}^d = \boldsymbol{\Pi}^\top \nabla_x^\top \boldsymbol{b}_x^d + \boldsymbol{\Pi}^\top \nabla_y^\top \boldsymbol{b}_y^d,$$

$$\text{where } \boldsymbol{A}_{\boldsymbol{\Pi}}^d \odot (\boldsymbol{A}_{\boldsymbol{\Pi}}^d)^\top = \left(\omega \boldsymbol{\Pi}^\top \boldsymbol{\Pi} + \boldsymbol{\Pi}^\top \nabla_x^\top \nabla_x \boldsymbol{\Pi} + \boldsymbol{\Pi}^\top \nabla_y^\top \nabla_y \boldsymbol{\Pi} + \phi \boldsymbol{\Pi}^\top (\nabla^2)^\top \nabla^2 \boldsymbol{\Pi}\right) \mathcal{D}^d,$$

$$\text{where } \boldsymbol{b}_x^d = \max\big(\nabla_x i^m, \nabla_x i^n\big) - \boldsymbol{\Pi}\nabla_x \mathcal{D}^s, \quad \boldsymbol{b}_y^d = \max\big(\nabla_y i^m, \nabla_y i^n\big) - \boldsymbol{\Pi}\nabla_y \mathcal{D}^s. \quad (15)$$

Analogously, the LMG mechanism $\mathcal{P}_{\mathcal{M}}\big(\hat{\boldsymbol{z}}_0^t, \{\boldsymbol{y}_i\}_{i=1}^2\big)$ can be formulated as:

$$\mathcal{P}_{\mathcal{M}}\big(\hat{\boldsymbol{z}}_0^t, \{\boldsymbol{y}_i\}_{i=1}^2\big) = \nabla_{\boldsymbol{z}_t}\mathcal{L}_{\text{Fusion}}\big(\boldsymbol{\Pi}(\mathcal{D}(\hat{\boldsymbol{z}}_0^t))\big) + \gamma_{\text{latent}}\nabla_{\boldsymbol{z}_t}\big\|\hat{\boldsymbol{z}}_0^t - \mathcal{E}(\boldsymbol{\Psi}(\max(i^m, i^n)))\big\|_2^2. \quad (16)$$

Here, the operator $\boldsymbol{\Psi}\big(\max(i^m, i^n)\big)$ can be represented by the following equation:

$$\boldsymbol{\Psi}\big(\max(i^m, i^n)\big) = \boldsymbol{\Pi}^\top(\max(i^m, i^n)) + \hat{\boldsymbol{z}}_0^t - \boldsymbol{\Pi}^\top \boldsymbol{\Pi}\hat{\boldsymbol{z}}_0^t. \quad (17)$$

**Additional implementation details and derivations are provided in Appendix A.**

## 5 Experiments

**Configuration.** We evaluate our method on three representative image fusion tasks: infrared-visible fusion (IVF), multi-exposure fusion (MEF), and multi-focus fusion (MFF). For IVF, 200 test samples from the LLVIP dataset [15] are used. MEF and MFF are evaluated following the testing setups of the MEFB [62] and MFFW [50] datasets, respectively. In addition, we conducted semantic

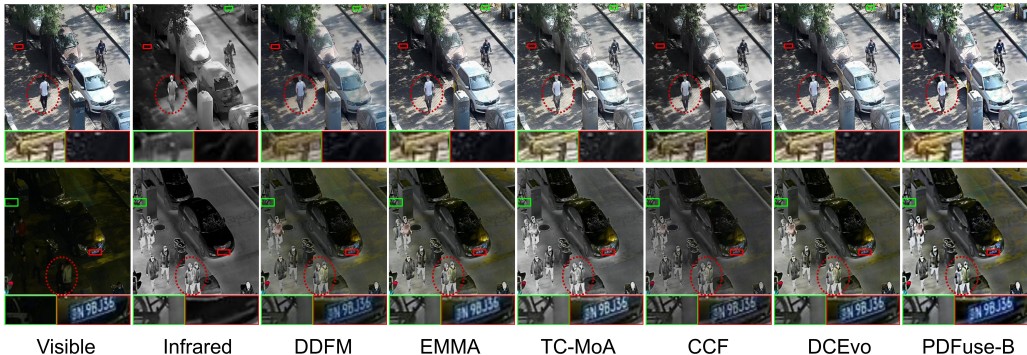

| Visible | Infrared | DDFM | EMMA | TC-MoA | CCF | DCEvo | PDFuse-B |

Figure 3: Qualitative comparison of PDFuse-B and state-of-the-art methods on the LLVIP dataset.

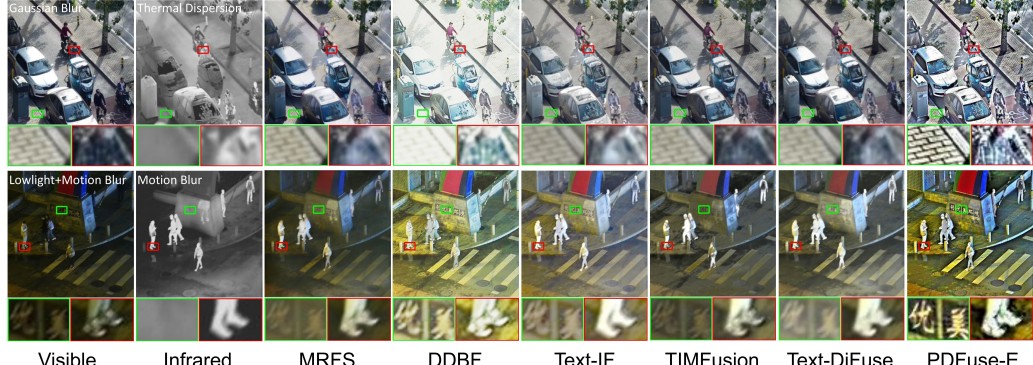

| Visible | Infrared | MRFS | DDBF | Text-IF | TIMFusion | Text-DiFuse | PDFuse-E |

Figure 4: Qualitative comparison on the LLVIP dataset under complex degradation scenarios between our enhanced PDFuse-E and state of the art degradation aware image fusion methods.

Table 1: Overall quantitative evaluation of infrared–visible image fusion on the LLVIP dataset using our PDFuse-B and PDFuse-E methods, with the upper subplot showing methods unable to handle degradation and the lower subplot showing degradation-handling methods. The **red**/**blue**/**green** indicates the best, runner-up and third best. **Rank** is the mean ranking across all metrics.

| Method | EI↑ | SF↑ | EN↑ | AG↑ | SD↑ | TE↓ | SCD↑ | $Q_{cb}$↑ | Rank↓ |
|---|---|---|---|---|---|---|---|---|---|
| **U2Fusion** (TPAMI'22) [48] | 28.253 | 6.628 | 6.447 | 2.481 | 29.286 | 8.436 | 0.659 | **0.455** | 7.0 |
| **DeFusion** (ECCV'22) [20] | 29.604 | 6.537 | 7.122 | 2.513 | 39.623 | 7.761 | 0.705 | 0.390 | 6.9 |
| **DDFM** (ICCV'23) [65] | 31.124 | 6.854 | 7.141 | 2.649 | 40.433 | 7.742 | **1.081** | 0.390 | 5.1 |
| **EMMA** (CVPR'24) [66] | 40.653 | 9.419 | 7.314 | **3.561** | **46.509** | 7.568 | **0.992** | **0.428** | **3.5** |
| **TC-MoA** (CVPR'24) [67] | **42.567** | **9.805** | **7.404** | **3.696** | **47.623** | **7.479** | 1.013 | 0.422 | **2.3** |
| **CCF** (NeurIPS'24) [3] | 31.342 | 7.513 | 7.045 | 2.708 | 39.255 | 7.838 | 0.977 | 0.394 | 5.9 |
| **DCEvo** (CVPR'25) [25] | **40.338** | **9.504** | **7.346** | 3.509 | 46.425 | **7.537** | 0.953 | 0.418 | 3.9 |
| **PDFuse-B** (Ours) | **48.519** | **11.028** | **7.427** | **4.221** | **47.931** | **7.457** | 0.981 | **0.437** | **1.5** |
| **DDBF** (CVPR'24) [57] | **55.108** | **13.082** | **7.336** | **4.806** | **48.308** | 7.547 | **1.021** | 0.413 | **2.4** |
| **Text-IF** (CVPR'24) [52] | **44.917** | **9.768** | 7.302 | **3.844** | 42.485 | 7.581 | **1.001** | **0.413** | 3.4 |
| **MRFS** (CVPR'24) [58] | 26.828 | 6.624 | 6.719 | 2.278 | 36.401 | 8.164 | 0.696 | 0.382 | 5.8 |
| **TIMFusion** (TPAMI'24) [26] | 34.206 | 7.363 | 6.818 | 2.910 | 36.079 | 8.653 | 0.482 | 0.397 | 5.3 |
| **Text-DiFuse** (NeurIPS'24) [56] | 38.744 | 8.711 | **7.417** | 3.255 | **52.255** | **7.466** | **1.255** | 0.403 | **2.9** |
| **PDFuse-E** (Ours) | **140.302** | **35.079** | **7.617** | **13.634** | **52.492** | **7.265** | 0.824 | **0.476** | **1.4** |

segmentation experiments on the FMB dataset [22] to verify the effectiveness of the fused images generated by our method in high-level vision tasks. All experiments ran on an NVIDIA GeForce RTX 3090 GPU and a 2.4 GHz Intel Xeon Silver 4210R CPU. Quantitative evaluation includes three reference-based metrics: Correlation-Based Quality ($Q_{cb}$) [29], Sum of the Correlations of Differences (SCD) [1], and Total Entropy (TE) [24], as well as five no-reference metrics: Edge Intensity (EI) [33], Spatial Frequency (SF) [7], Entropy (EN) [35], Average Gradient (AG) [6], and Standard Deviation (SD) [34].

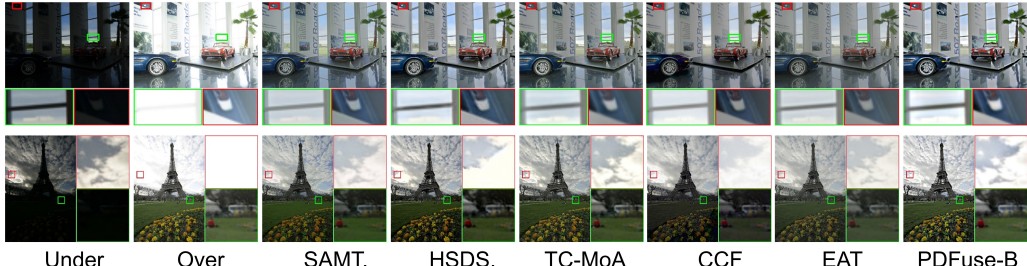

| Under | Over | SAMT. | HSDS. | TC-MoA | CCF | EAT | PDFuse-B |

Figure 5: Qualitative comparison of PDFuse-B and state-of-the-art methods on the MEFB dataset.

**Evaluation on infrared-visible fusion.** We evaluate our approach on the LLVIP dataset [15] by generating paired clean and degraded test images via degradation removal [5] and addition operations. To ensure a fair comparison, we form two experimental groups. In the first group, our base model PDFuse-B is compared with four state-of-the-art general fusion methods (U2Fusion [48], DeFusion [20], TC-MoA [67] and CCF [3]) and three task-specific multi-modal methods (DDFM [65], EMMA [66] and DCEvo [25]), none of which can handle degradation. The qualitative results, encompassing both daytime and nighttime scenarios as demonstrated in Fig. 3, along with quantitative comparisons summarized in Table 1, reveal that our method achieves superior preservation of texture and salient object information. In particular, according to the qualitative analysis, our method exhibits superior infrared salience in pedestrians and effectively preserves fine details in both infrared and visible textures (e.g., bicycle and tree trunk textures), and it attains the highest average ranking across all evaluation metrics.

In the second group, we evaluate our enhanced variant, PDFuse-E, which incorporates a degradation operator library, against five leading degradation-aware fusion methods (DDBF [57], MRFS [58], Text-IF [52], TIMFusion [26], Text-DiFuse [56]). Fig. 4 illustrates our method's effectiveness in removing degradation and retaining detail. As shown in the lower half of Table 1, PDFuse-E outperforms current state-of-the-art techniques in texture, contrast, and structural metrics and secures the highest Rank.

Table 2: Quantitative comparison of multi-exposure image fusion methods on the MEFB dataset. The **red**/**blue**/**green** denotes first, second and third best. **Rank** is the mean ranking across all metrics.

| Method | EI↑ | SF↑ | EN↑ | AG↑ | SD↑ | TE↓ | SCD↑ | Q_cb↑ | Rank↓ |
|---|---|---|---|---|---|---|---|---|---|
| **U2Fusion** (TPAMI'22) [48] | 49.381 | 14.503 | 6.767 | 4.393 | 46.789 | 5.185 | 5.515 | 0.446 | 6.9 |
| **DeFusion** (ECCV'22) [20] | 52.704 | 15.319 | 6.799 | 4.570 | 52.734 | 5.150 | 0.275 | 0.370 | 6.5 |
| **HSDS-MEF** (AAAI'24) [45] | 64.217 | 19.379 | 7.136 | 5.951 | 55.440 | 4.813 | 0.859 | 0.449 | 2.4 |
| **TC-MoA** (CVPR'24) [67] | 56.439 | 15.782 | 7.093 | 5.049 | 49.852 | 4.856 | 0.538 | 0.450 | 4.4 |
| **CCF** (NeurIPS'24) [3] | 51.933 | 17.501 | 7.134 | 4.855 | 60.174 | 4.815 | 1.163 | 0.406 | 3.8 |
| **SAMT-MEF** (InF'24) [14] | 55.097 | 17.518 | 7.046 | 5.146 | 50.824 | 4.904 | 0.635 | 0.450 | 3.9 |
| **EAT** (TMM'25) [42] | 45.237 | 13.366 | 6.916 | 4.055 | 47.096 | 5.033 | 0.423 | 0.428 | 7.0 |
| **PDFuse-B** (Ours) | 69.772 | 20.695 | 7.271 | 6.316 | 60.463 | 4.678 | 0.774 | 0.457 | 1.3 |

**Evaluation on multi-exposure fusion.** We compare our method, PDFuse-B, against four general-purpose image fusion techniques (U2Fusion [48], DeFusion [20], TC-MoA [67], and CCF [3]) and three task-specific multi-exposure fusion methods (SAMT-MEF [14], HSDS-MEF [45] and EAT [42]) as competitors. As shown in Fig. 5, our approach delivers the most consistent global exposure and color while offering clear advantages in texture, contrast, and color saturation. Our method shows a significant advantage in overall exposure levels and color contrast compared to other methods, and the highlighted regions further confirm its superiority in preserving textures across images with different exposures (e.g., windows, posters, clouds). The superior quantitative results in Table 2 further confirm these improvements.

**Evaluation on multi-focus fusion.** In addition to the four general-purpose fusion methods, we compare our approach with two leading multi-focus fusion techniques, ZMFF [13] and DB-MFIF [60]. Fig. 6 demonstrates that our method more effectively preserves texture and combines details from images with different focus regions. In particular, our approach preserves textures across varying

Table 3: Quantitative comparison of multi-focus image fusion methods on the MFFW dataset. The red/blue/green denotes first, second and third best. **Rank** is the mean ranking across all metrics.

| Method | EI↑ | SF↑ | EN↑ | AG↑ | SD↑ | TE↓ | SCD↑ | Q_cb↑ | Rank↓ |
|---|---|---|---|---|---|---|---|---|---|
| **U2Fusion** (TPAMI'22) [20] | 72.763 | 17.152 | 6.953 | 6.595 | 48.967 | 7.312 | 0.465 | 0.586 | 5.8 |
| **DeFusion** (ECCV'22) [20] | 57.015 | 12.937 | 7.120 | 5.010 | 51.224 | 7.146 | 0.410 | 0.556 | 6.5 |
| **ZMFF** (InF'23) [13] | 85.799 | 22.175 | 7.188 | 8.026 | 53.682 | 7.077 | 0.500 | 0.671 | 3.4 |
| **TC-MoA** (CVPR'24) [67] | 74.781 | 18.620 | 7.184 | 6.805 | 54.094 | 7.081 | 0.719 | 0.650 | 4.0 |
| **CCF** (NeurIPS'24) [3] | 65.660 | 16.613 | 7.261 | 5.998 | 59.329 | 7.004 | 1.330 | 0.539 | 4.0 |
| **DB-MFIF** (TMM'24) [60] | 87.087 | 24.045 | 7.221 | 8.321 | 55.158 | 7.045 | 0.677 | 0.662 | 2.6 |
| **PDFuse-B** (Ours) | 93.756 | 22.523 | 7.278 | 8.463 | 60.445 | 6.990 | 0.894 | 0.576 | 1.8 |

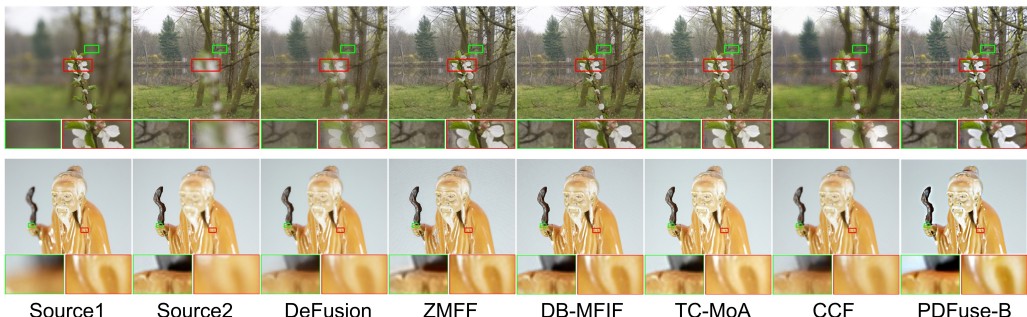

Figure 6: Qualitative comparison of PDFuse-B and state-of-the-art methods on the MFFW dataset.

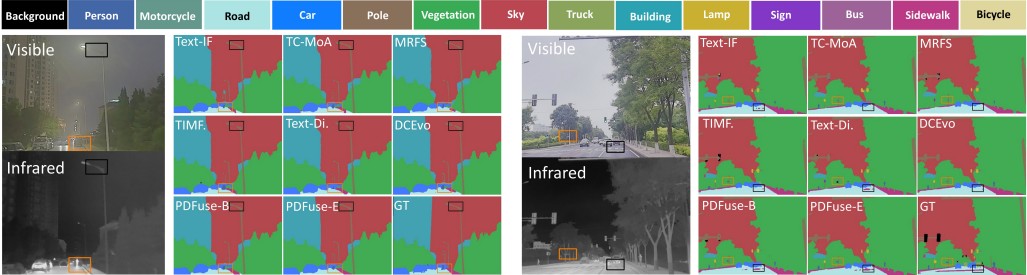

Figure 7: Qualitative comparison of semantic segmentation performance on the FMB dataset.

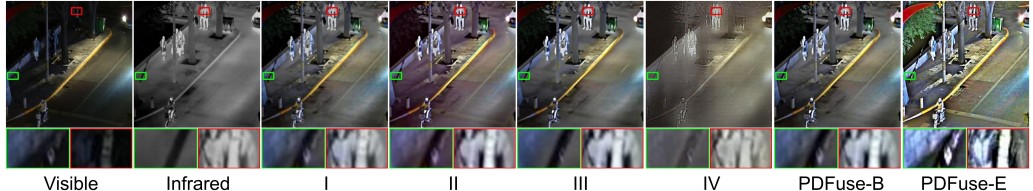

Figure 8: A qualitative comparison for the ablation study on the LLVIP dataset.

focus depths while maintaining high overall image contrast. When compared to the diffusion-based CCF [3], our method achieves markedly higher texture fidelity. These qualitative advantages are corroborated by the quantitative results in Table 3, which show superior performance across multiple multi-focus fusion metrics.

**Semantic verification on segmentation.** We retrained SegNeXt-B [8] on the fusion outputs of all methods from the FMB training set and evaluated on the corresponding test set [22]; qualitative comparisons are presented in Fig. 7 and quantitative results in Table 4, where our enhanced method PDFuse-E attains the highest segmentation mIoU and mAcc.

**Ablation studies.** We performed an ablation study on the LLVIP dataset to assess the impact of each core component in our framework, including **(I)** the LMG mechanism, **(II)** the first projection stage of the MICP mechanism, **(III)** the second projection stage of the MICP mechanism, and **(IV)**

Table 4: Quantitative comparison of semantic segmentation performance on fused images from the FMB dataset. The **red**/**blue**/**green** indicates the best, runner-up and third best.

| Method | Person | SKy | Lamp | Car | Bus | Truck | Motor | Pole | mAcc | mIoU |
|--------|--------|-----|------|-----|-----|-------|-------|------|------|------|
| DDBF | 62.97 | 93.73 | 33.11 | 81.60 | 35.00 | 49.38 | 45.81 | 46.61 | 68.12 | 59.89 |
| EMMA | 68.27 | 93.47 | 44.43 | 81.71 | 34.87 | 47.18 | 49.55 | 47.92 | 70.69 | 61.87 |
| Text-IF | 67.08 | 94.04 | 40.00 | 81.77 | 28.00 | 48.48 | 47.47 | 48.52 | 69.54 | 61.15 |
| TC-MoA | 67.34 | 94.11 | 43.02 | 80.90 | 41.29 | 43.15 | 44.61 | 47.31 | 69.69 | 61.67 |
| MRFS | 68.37 | 94.09 | 36.95 | 84.02 | 56.55 | 49.20 | 46.27 | 47.12 | 71.16 | 63.00 |
| TIMFusion | 66.04 | 93.72 | 41.43 | 83.62 | 50.71 | 46.07 | 44.41 | 45.70 | 70.72 | 62.25 |
| Text-DiFuse | 67.79 | 94.12 | 37.44 | 83.04 | 62.96 | 39.16 | 45.24 | 46.56 | 70.98 | 62.61 |
| DCEvo | 69.44 | 93.81 | 41.42 | 83.09 | 62.65 | 28.71 | 43.54 | 45.45 | 70.89 | 62.53 |
| PDFuse-B | 69.06 | 94.07 | 39.23 | 82.22 | 43.77 | 45.30 | 49.72 | 48.25 | 70.97 | 62.54 |
| PDFuse-E | 66.88 | 93.80 | 45.51 | 83.83 | 57.99 | 48.15 | 47.09 | 47.65 | 71.77 | 63.70 |

Table 5: Quantitative evaluation of the ablation experiments on the LLVIP dataset.

| Method | MICP. | LMG. | Opr. | EI↑ | SF↑ | EN↑ | AG↑ | SD↑ | TE↓ | SCD↑ | $Q_{cb}$↑ |
|--------|-------|------|------|-----|-----|-----|-----|-----|-----|------|-----------|
| I | ✓ | ✗ | ✗ | 45.26 | 10.01 | 7.394 | 3.941 | 46.71 | 7.489 | 0.975 | 0.443 |
| II | ✗/st.1 | ✓ | ✗ | 47.72 | 10.59 | 7.408 | 4.132 | 47.36 | 7.475 | **0.987** | 0.432 |
| III | ✗/st.2 | ✓ | ✗ | 42.54 | 10.06 | 7.385 | 3.661 | 47.11 | 7.498 | 0.901 | 0.406 |
| IV | ✗ | ✓ | ✗ | 33.76 | 8.413 | 7.155 | 2.892 | 40.85 | 7.718 | 0.678 | 0.316 |
| PDFuse-B | ✓ | ✓ | ✗ | 48.52 | 11.03 | 7.427 | 4.221 | 47.93 | 7.457 | 0.981 | 0.437 |
| PDFuse-E | ✓ | ✓ | ✓ | **140.3** | **35.08** | **7.617** | **13.63** | **52.49** | **7.265** | 0.824 | **0.476** |

the entire MICP mechanism; qualitative results are presented in Fig. 8 and quantitative comparisons are reported in Table 5.

The results reveal that removing the first projection stage of MICP causes color distortion because this stage injects the coarse structural component $\mathcal{D}^s$ into the diffusion process, and without it low-frequency and color information cannot be effectively preserved. Removing the second projection stage leads to significant texture loss, since it injects the fine-detail component $\mathcal{D}^d$. When the entire MICP mechanism is removed, relying solely on the LMG mechanism fails to maintain information fidelity and results in poor fusion quality. Conversely, if the LMG mechanism is removed while MICP remains, the MICP mechanism still supports structural and texture preservation, but the fusion exhibits reduced contrast and darker tones compared to PDFuse-B (which includes LMG), due to the lack of manifold-preservation guidance and consequently limited use of generative priors.

Finally, PDFuse-E, which integrates the interference operator set, outperforms PDFuse-B by leveraging diffusion priors to address inverse degradation problems (e.g., low light and blur), thereby producing better overall visual quality. These findings demonstrate that the full configuration delivers the best overall performance.

# 6 Conclusion

This paper proposes PDFuse, a robust, training-free image fusion framework built on pre-trained latent diffusion models with projection–manifold regularization. PDFuse formulates fusion as a multi-source regularized latent diffusion process and employs two complementary regularization mechanisms to ensure high fidelity and effective use of diffusion priors. It also integrates a canonical interference operator set into these mechanisms, allowing the model to address diverse degradations without clear-image supervision. Extensive experiments demonstrate that PDFuse outperforms current methods in both fusion quality and preservation of semantic attributes.

# 7 Acknowledgement

This work was supported by the NSFC (62506268, 62276192) and the Natural Science Foundation of Jiangsu Province (BK20250454).

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

# Appendix

## A  More details of PDFuse

To further enhance the clarity and completeness of our discussion, we provide additional architectural details of the proposed PDFuse framework.

### A.1  More details of the MICP mechanism

**The fine detail projection.**  The Sobel operator $\nabla$ used for extracting texture features can be decomposed into a horizontal gradient operator $\nabla_x$ and a vertical gradient operator $\nabla_y$. Accordingly, the detail-constraint term in Eq. (7) can be reformulated as

$$\min_{\mathcal{D}^d} \left( \underbrace{\left\| \nabla_x(\mathcal{D}^s + \mathcal{D}^d) - \max\left(\nabla_x i^m, \nabla_x i^n\right) \right\|_2^2 + \left\| \nabla_y(\mathcal{D}^s + \mathcal{D}^d) - \max\left(\nabla_y i^m, \nabla_y i^n\right) \right\|_2^2}_{\text{detail-constraint term}} \right). \tag{18}$$

Taking the derivative of Eq. (7) with respect to $\mathcal{D}^d$ yields:

$$\frac{\partial \mathcal{L}}{\partial \mathcal{D}^d} = 2 \left[ \nabla_x^\top \left( \nabla_x(\mathcal{D}^s + \mathcal{D}^d) - \mathcal{T}_x \right) + \nabla_y^\top \left( \nabla_y(\mathcal{D}^s + \mathcal{D}^d) - \mathcal{T}_y \right) + \omega \mathcal{D}^d + \phi(\nabla^2)^\top (\nabla^2) \mathcal{D}^d \right]. \tag{19}$$

Specifically, we define $\mathcal{T}_x = \max(\nabla_x i^m, \nabla_x i^n)$ and $\mathcal{T}_y = \max(\nabla_y i^m, \nabla_y i^n)$. By setting $\frac{\partial \mathcal{L}}{\partial \mathcal{D}^d} = 0$ and rearranging the terms, we obtain:

$$\left[ \nabla_x^\top \nabla_x + \nabla_y^\top \nabla_y + \omega \mathbf{I} + \phi(\nabla^2)^\top (\nabla^2) \right] \mathcal{D}^d = \nabla_x^\top (\mathcal{T}_x - \nabla_x \mathcal{D}^s) + \nabla_y^\top (\mathcal{T}_y - \nabla_y \mathcal{D}^s). \tag{20}$$

This result is consistent with Eq. (10) in the main text.

**Conjugate gradient method.**  The two-stage projection system is fully solvable via the Conjugate Gradient (CG) method. In the first-stage projection, the coefficient matrix reduces to the identity operator $\mathbf{I}$, which is trivially symmetric positive definite (SPD) since $\langle x, \mathbf{I}x \rangle = \|x\|_2^2 > 0$ for all $x \neq 0$. For the second-stage projection with composite operator:

$$\mathbf{A}^d \odot (\mathbf{A}^d)^\top = \underbrace{\nabla_x^\top \nabla_x + \nabla_y^\top \nabla_y}_{\text{gradient terms}} + \underbrace{\omega \mathbf{I}}_{\text{regularization}} + \underbrace{\phi(\nabla^2)^\top \nabla^2}_{\text{high-order term}}, \tag{21}$$

is solvable via the Conjugate Gradient (CG) method [38] since $\mathbf{A}^d \odot (\mathbf{A}^d)^\top$ satisfies the required symmetry and positive definiteness. Symmetry arises from: 1) Adjoint Sobel operators $\nabla_x^\top = \nabla_x^T$ and $\nabla_y^\top = \nabla_y^T$, 2) Self-adjoint Laplacian $\nabla^2 = \nabla_x^\top \nabla_x + \nabla_y^\top \nabla_y$, and 3) Explicit symmetry of $\omega \mathbf{I}$. Positive definiteness follows from:

$$\langle x, \mathbf{A}^d \odot (\mathbf{A}^d)^\top x \rangle = \underbrace{\|\nabla_x x\|_2^2 + \|\nabla_y x\|_2^2}_{\geq 0} + \underbrace{\omega \|x\|_2^2}_{>0} + \underbrace{\phi \|\nabla^2 x\|_2^2}_{\geq 0} > 0, \ \forall x \neq 0, \tag{22}$$

where the strict inequality holds due to the $\omega \mathbf{I}$ term. The CG method therefore guarantees convergence to the unique solution within finite iterations.

### A.2  More details of alternating optimization

We collected a series of intermediate diffusion results under the regularization-constrained diffusion process, $\mathcal{D}(\hat{z}_0^t)$, which allows for an intuitive visualization of the effects imposed by different regularization mechanisms, as demonstrated in Fig. 9. From these results, it is evident that the MICP mechanism produces a pronounced enhancement of multi-source information, whereas the LMG mechanism yields an overall improvement in visual quality.

### A.3  More details of interference operator set

The interference operator set $\{\mathbf{\Pi}_i\}_{i=1}^N$ is defined as a library of degradation operators, which includes a collection of representative interference models. These operators are integrated into the bidirectional inference mechanism of our framework to address potential disturbances that may arise during the fusion process.

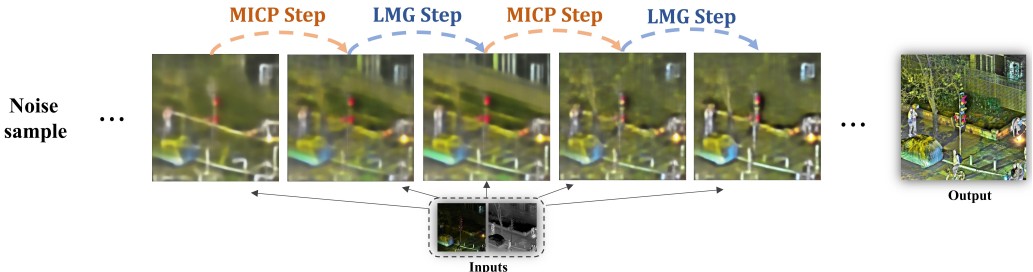

Figure 9: An intuitive visualization of the effects imposed by different regularization mechanisms.

**Implementation details of representative degradation operators.** For degradations caused by low-light conditions and overexposure, we adopt the illumination decomposition theory based on the Retinex model, where the observed image is represented as $i^E = i^L/L$. To estimate the degraded illumination component, we follow the approach proposed in [9]. Specifically, for handling both low-light and overexposed regions, we leverage forward and inverse illumination estimation strategies as described in [61]. The resulting illumination map $L$ is then incorporated into our framework to guide the optimization of illumination-aware fusion. Other degradation types, including Gaussian blur, motion blur, and thermal diffusion in infrared images, are implemented via convolutional operations. By constructing both the degradation operators and their corresponding transpose (adjoint) operators, these degradation processes can be universally integrated into our proposed bidirectional inference mechanism, thereby assisting the diffusion-based reasoning in a more robust manner.

**Integrated strategy.** Given a set of multi-source image inputs $\{y_i\}_{i=1}^2$, when each image is subject to different degradations $\mathbf{\Pi}_i$ and $\mathbf{\Pi}_j$ drawn from a predefined degradation operator set $\{\mathbf{\Pi}_i\}_{i=1}^N$, directly applying the basic projection-manifold regularization inevitably results in the fused image inheriting the degradations from the source images, which is an undesirable outcome. Therefore, we further revise the formulation in Eq. (7) as follows:

$$\mathcal{L}_{\text{Fusion}}(\mathbf{\Pi}\mathcal{D}(\hat{z}_0^t)) = \gamma_{\text{int}} \left\| \mathbf{\Pi}\mathcal{D}(\hat{z}_0^t) - \max(i^m, i^n) \right\|_2^2 + \gamma_{\text{grad}} \left\| \nabla\left(\mathbf{\Pi}\mathcal{D}(\hat{z}_0^t)\right) - \max\left(\nabla i^m, \nabla i^n\right) \right\|_2^2. \tag{23}$$

In this context, $\mathbf{\Pi}$ denotes a weighted combination of $\mathbf{\Pi}_i$ and $\mathbf{\Pi}_j$, where the weighting coefficients are determined by the fusion contributions of the respective source images. When the source images undergo different types of degradation, the consistency loss between $\mathbf{\Pi}\hat{z}_0^t$ and the multi-source images is established by treating $\mathbf{\Pi}$ as a fused operator derived from $\mathbf{\Pi}_i$ and $\mathbf{\Pi}_j$. The fusion coefficients are determined by the spatial weights of the multi-source images during each stage of regularization.

Taking the first-stage projection of the MICP mechanism in the VIS task as an example, in order to preserve the saliency of the infrared image and the majority of scene details from the visible image, the fusion is typically performed using the $\max(\cdot)$ operator. In this case, the spatial fusion weights $\text{coeff}_m$ and $\text{coeff}_n$ satisfy the relation:

$$\text{coeff}_m \cdot i^m + \text{coeff}_n \cdot i^n = \max(i^m, i^n). \tag{24}$$

Accordingly, the fusion of the interference operator sets $\mathbf{\Pi}_i$ and $\mathbf{\Pi}_j$ also follows this coefficient scheme. In the second stage of the MICP mechanism, the fusion coefficients for the interference operator sets are similarly derived based on the fusion weights computed over Sobel gradients. Each component of the LMG mechanism adheres to similar principles. Based on the same derivation process described earlier, the resulting projection-manifold regularization formulations correspond to Eq. (14), Eq. (15) and Eq. (16).

## B Explanation on VAE fidelity

VAE does exhibit significant information bias and loss when applied to high-quality visible images with very dense and regular textures. However, the loss introduced by VAE is typically concentrated in its high-frequency components.We perform a 2D FFT on both the original image and the VAE-encoded and decoded reconstructed image, generating a radial frequency grid followed by frequency band partitioning, which is used to study the distribution of MSE losses from VAE encoding and decoding across different normalized frequency bands on the LLVIP dataset, as shown in Table 6.

Table 6: The distribution of MSE losses from VAE encoding and decoding across different normalized frequency bands on the LLVIP dataset.

| Frequency Range (cycles/pixel) | 0.00-0.025 | 0.025-0.050 | 0.050-0.100 | 0.100-0.250 | 0.250-0.500 |
|---|---|---|---|---|---|
| MSE Loss | 0.000000 | 0.000001 | 0.000003 | 0.000062 | 0.001003 |

It is clear that VAE's loss and reconstruction bias are concentrated in the high frequencies. To address this challenge, we adopted a high-frequency residual injection strategy. Specifically, in our MICP mechanism, the regularized projection captures the coarse structural component $\mathcal{D}^s$ and the fine detail component $\mathcal{D}^d$ from the multi-source image set $i^m, i^n$. Then, a high-frequency filter $\mathcal{G}(\cdot)$ is applied to remove the high-frequency components from the projection result as in Eq. (11). In other words, the high-frequency information is kept as $R$ rather than encoded into the latent space. Moreover, $R$ is iteratively refined via projection regularization of $\hat{z}_0^{t-1}$ and finally injected as a residual after the process completes. Additionally, we further conduct an ablation study on the high-frequency filtering and residual structure. The MICP mechanism projection result is directly encoded by VAE into the latent space (without pre-reserving high-frequency information), and the quantitative comparison is shown in Table 7.

Table 7: Ablation comparison of high-frequency information skip residual injection.

| Methods | EI↑ | SF↑ | AG↑ | EN↑ | SD↑ | TE↓ | SCD↑ | $\mathbf{Q_{cb}}$↑ |
|---|---|---|---|---|---|---|---|---|
| **wo / (filtering and residual)** | 39.391 | 8.070 | 7.359 | 3.371 | 45.077 | 7.528 | 0.913 | 0.418 |
| **PDFuse** | **48.519** | **11.028** | **7.427** | **4.221** | **47.931** | **7.457** | **0.981** | **0.437** |

Based on the ablation results, the model variant lacking high-frequency filtering and residual components exhibits a significant loss in edge and texture details. These results demonstrate the effectiveness of our adopted high-frequency residual injection strategy.

## C  Algorithm flowchart

This figure presents the algorithmic flowchart of our proposed method, the projection-manifold regularized latent diffusion framework. The flowchart is divided into two variants: the basic version (PDFuse-B) and the enhanced version (PDFuse-E).

---

**Algorithm 1** PDFuse-B

**Require:** $T, \{\alpha\}_{t=1}^T, \mathcal{D}, i^m, i^n$
1: Sample $z_T \sim \mathcal{N}(0, I)$
2: **for** $i = T$ **to** 1 **do**
3:     $\hat{\epsilon}_\theta \leftarrow s_\theta(z_t, t, c)$
4:     $\hat{z}_0^t \leftarrow \frac{z_t - \sqrt{1-\bar{\alpha}_t}\,\hat{\epsilon}_\theta}{\sqrt{\bar{\alpha}_t}}$
5:     **if** MICP step **then**
6:         $\mathcal{D}^s \leftarrow$ Eq. (8)$^\dagger$
7:         $\mathcal{D}^d \leftarrow$ Eq. (10)$^\dagger$
8:         $\mathcal{R} \leftarrow (\mathbf{I} - \mathcal{G})(\mathcal{D}^s + \mathcal{D}^d)$
9:         $\mathcal{F}_\theta(\hat{z}_0^t, \{y_i\}_{i=1}^2) \leftarrow$ Eq. (11)$^\dagger$
10:         $\mathcal{P}_\mathcal{M}(\hat{z}_0^t, \{y_i\}_{i=1}^2) \leftarrow 0$
11:         $z_{t-1} \leftarrow$ Eq. (6)$^\dagger$
12:     **else if** LGM step **then**
13:         $\mathcal{F}_\theta(\hat{z}_0^t, \{y_i\}_{i=1}^2) \leftarrow \hat{z}_0^t$
14:         $\mathcal{P}_\mathcal{M}(\hat{z}_0^t, \{y_i\}_{i=1}^2) \leftarrow$ Eq. (12)$^\dagger$
15:     **end if**
16: **end for**
17: **return** $\mathcal{D}(z_{t-1}) + \mathcal{R}$

---

**Algorithm 2** PDFuse-E

**Require:** $T, \{\alpha\}_{t=1}^T, \mathcal{D}, i^m, i^n$
1: Sample $z_T \sim \mathcal{N}(0, I)$
2: **for** $i = T$ **to** 1 **do**
3:     $\hat{\epsilon}_\theta \leftarrow s_\theta(z_t, t, c)$
4:     $\hat{z}_0^t \leftarrow \frac{z_t - \sqrt{1-\bar{\alpha}_t}\,\hat{\epsilon}_\theta}{\sqrt{\bar{\alpha}_t}}$
5:     **if** MICP step **then**
6:         $\mathcal{D}^s \leftarrow$ Eq. (14)$^\dagger$
7:         $\mathcal{D}^d \leftarrow$ Eq. (15)$^\dagger$
8:         $\mathcal{R} \leftarrow (\mathbf{I} - \mathcal{G})(\mathcal{D}^s + \mathcal{D}^d)$
9:         $\mathcal{F}_\theta(\hat{z}_0^t, \{y_i\}_{i=1}^2) \leftarrow$ Eq. (11)$^\dagger$
10:         $\mathcal{P}_\mathcal{M}(\hat{z}_0^t, \{y_i\}_{i=1}^2) \leftarrow 0$
11:         $z_{t-1} \leftarrow$ Eq. (6)$^\dagger$
12:     **else if** LGM step **then**
13:         $\mathcal{F}_\theta(\hat{z}_0^t, \{y_i\}_{i=1}^2) \leftarrow \hat{z}_0^t$
14:         $\mathcal{P}_\mathcal{M}(\hat{z}_0^t, \{y_i\}_{i=1}^2) \leftarrow$ Eq. (16)$^\dagger$
15:     **end if**
16: **end for**
17: **return** $\mathcal{D}(z_{t-1}) + \mathcal{R}$

---

# D    Support image fusion with more than two source images

Since our method formulates the fusion process as a regularized latent diffusion conditioned on multi-source images, it does not impose an explicit limit on the number of inputs. Consequently, our approach can process an arbitrary number of source images in a single pass with negligible additional computational or temporal overhead. We validated this capability on the SICE multi-exposure dataset [2] by inputting three images with different exposure levels. It is worth emphasizing that other methods can only process two images at a time, so they perform multi-exposure fusion of three images in two separate stages via sequential pairing. In contrast, our method achieves this in a single pass. The quantitative results are presented in Table 8.

From the experimental results, our approach achieves leading performance on most evaluation metrics, demonstrating that our regularization mechanism can be generally applied to fuse three or more source images. In addition, we further analyzed how the runtime and the Edge Intensity (EI) of the fused images produced by our method, PDFuse, change as the number of input images with different exposure settings varies. The quantitative results are presented in Table 9.

Table 8: Quantitative comparison of multi-exposure image fusion methods with 3 exposure inputs on the SICE dataset [2]. The **red**/**blue**/**green** denotes first, second and third best.

| Methods | EI↑ | SF↑ | EN↑ | AG↑ | SD↑ | TE↓ | SCD↑ | Q$_{cb}$↑ |
|---|---|---|---|---|---|---|---|---|
| U2Fusion | 85.165 | 22.561 | 7.225 | 8.048 | 54.565 | 13.255 | 0.716 | **0.500** |
| DeFusion | 67.214 | 17.257 | 7.279 | 6.206 | 52.841 | 13.201 | 0.550 | 0.357 |
| HSDS-MEF | **95.637** | **29.464** | 6.994 | **9.367** | **73.821** | 13.486 | **2.989** | 0.351 |
| TC-MoA | **89.575** | 24.851 | **7.532** | 8.611 | 57.462 | **12.947** | 0.959 | **0.466** |
| CCF | 86.120 | 28.723 | **7.614** | 8.885 | **70.408** | **12.866** | **3.575** | 0.461 |
| SAMT-MEF | 89.485 | **29.514** | 7.425 | **9.378** | 58.071 | 13.055 | 1.225 | 0.430 |
| EAT | 36.849 | 9.384 | 6.501 | 3.146 | 40.089 | 13.937 | 1.700 | 0.347 |
| PDFuse-B | **112.422** | **30.975** | **7.708** | **10.930** | 68.581 | **12.771** | **2.349** | **0.489** |

Table 9: Comparison of Edge Intensity (EI) and Runtime versus the Number of Input Exposure Sequences on the SICE Dataset.

| Input Sequence Counts | 2 | 3 | 4 | 5 | 6 | 7 | 8 | 9 |
|---|---|---|---|---|---|---|---|---|
| **Edge Intensity (EI)**↑ | 95.155 | 112.422 | 117.805 | 119.518 | 121.840 | 121.855 | 123.009 | **124.45** |
| **Runtime (s)** | 18.08 | 18.15 | 18.03 | 18.24 | 18.33 | 18.19 | 18.30 | 18.23 |

It can be observed that, for the same scene, as the number of input images with different exposure settings increases, the scene information texture (EI) of the fused images gradually improves, while the runtime remains essentially unchanged.

# E    More comprehensive experimental evaluations

To conduct more extensive experimental evaluations and further demonstrate our method's generality, we carried out additional assessments on larger benchmark datasets for infrared-visible image fusion, multi-exposure image fusion, and multi-focus image fusion.

## E.1    More evaluations for infrared-visible fusion

We conducted additional evaluations on four widely used infrared–visible fusion benchmarks: the TNO, FMB [23], MSRS [40], and RoadScene [46] datasets. Two experimental groups were established. In the first group, our base model (PDFuse-B) was compared with current state-of-the-art fusion methods. In the second group, our enhanced model (PDFuse-E), which integrates degradation operators, was evaluated against leading fusion methods capable of handling degraded inputs. The TNO dataset comprises 42 image pairs from diverse scenes; MSRS employs a 300-image test set; and RoadScene and FMB were each assessed on randomly sampled subsets of 200 and 150 image pairs, respectively. Quantitative results for these four datasets are shown in Fig. 14, Fig. 15, Fig. 16, and Fig. 17. PDFuse-E consistently improves target saliency, preserves fine texture details, and enhances overall visual quality. Furthermore, numerical comparisons in Table 11 and Table 10 demonstrate that both PDFuse-B and PDFuse-E achieve the highest mean scores across all evaluated metrics.

## E.2    More evaluations for multi-exposure image fusion

Table 10: Overall quantitative evaluation of infrared–visible image fusion on the MSRS dataset and RoadScene Dataset using our PDFuse-B and PDFuse-E methods, with the upper subplot showing methods unable to handle degradation and the lower subplot showing degradation-handling methods. The red/blue/green indicates the best, runner-up and third best. **Rank** is the mean ranking across all metrics.

| Method | MSRS Dataset | | | | | | RoadScene Dataset | | | | | | Rank↓ |
|---|---|---|---|---|---|---|---|---|---|---|---|---|---|
| | EI↑ | SF↑ | EN↑ | AG↑ | SD↑ | TE↓ | EI↑ | SF↑ | EN↑ | AG↑ | SD↑ | TE↓ | |
| **U2Fusion** (TPAMI'22) | 23.306 | 6.838 | 4.919 | 2.093 | 19.753 | 6.799 | 56.238 | 11.783 | 6.843 | 4.801 | 32.461 | 7.438 | 7.3 |
| **DeFusion** (ECCV'22) | 28.744 | 7.928 | 6.355 | 2.535 | 36.603 | 5.362 | 42.430 | 8.607 | 6.898 | 3.390 | 36.378 | 7.383 | 6.4 |
| **DDFM** (ICCV'23) | 25.981 | 6.942 | 6.074 | 2.330 | 28.055 | 5.644 | 51.476 | 11.555 | 7.189 | 4.494 | 41.711 | 7.093 | 6.1 |
| **EMMA** (CVPR'24) | 38.692 | 10.818 | 6.614 | 3.462 | 42.959 | 5.104 | 72.728 | 16.376 | 7.446 | 6.298 | 56.223 | 6.835 | 2.0 |
| **TC-MoA** (CVPR'24) | 39.342 | 10.875 | 6.600 | 3.562 | 39.557 | 5.118 | 60.408 | 12.089 | 7.142 | 4.973 | 40.554 | 7.139 | 3.8 |
| **CCF** (NeurIPS'24) | 27.831 | 8.201 | 6.063 | 2.568 | 27.248 | 5.655 | 53.424 | 13.076 | 7.291 | 4.640 | 46.030 | 6.990 | 5.0 |
| **DCEvo** (CVPR'25) | 38.405 | 10.808 | 6.519 | 3.451 | 41.519 | 5.199 | 65.582 | 14.940 | 7.139 | 5.553 | 44.560 | 7.142 | 4.0 |
| **PDFuse-B** (Ours) | 44.223 | 12.630 | 6.703 | 3.959 | 43.429 | 5.014 | 76.644 | 16.670 | 7.210 | 6.535 | 46.422 | 7.071 | 1.4 |
| **DDBF** (CVPR'24) | 51.530 | 13.959 | 6.945 | 4.523 | 51.864 | 4.772 | 45.102 | 10.307 | 5.834 | 3.521 | 23.518 | 8.447 | 4.4 |
| **Text-IF** (CVPR'24) | 38.081 | 9.473 | 6.967 | 3.332 | 42.122 | 4.750 | 74.194 | 16.701 | 7.422 | 6.496 | 50.543 | 6.859 | 3.1 |
| **MRFS** (CVPR'24) | 27.905 | 8.456 | 6.411 | 2.405 | 37.081 | 5.307 | 51.052 | 11.096 | 7.386 | 4.226 | 53.377 | 6.896 | 4.2 |
| **TIMFusion** (TPAMI'24) | 41.059 | 10.943 | 6.957 | 3.567 | 42.832 | 4.760 | 47.159 | 10.562 | 6.994 | 3.879 | 40.723 | 7.287 | 4.0 |
| **Text-DiFuse** (NeurIPS'24) | 39.538 | 10.740 | 7.082 | 3.413 | 52.621 | 4.635 | 45.382 | 9.982 | 7.310 | 3.714 | 50.807 | 6.971 | 3.4 |
| **PDFuse-E** (Ours) | 83.659 | 20.623 | 7.363 | 7.484 | 49.770 | 4.355 | 95.914 | 22.383 | 7.238 | 8.359 | 47.386 | 7.043 | 1.9 |

Table 11: Overall quantitative evaluation of infrared–visible image fusion on the TNO dataset and FMB Dataset using our PDFuse-B and PDFuse-E methods, with the upper subplot showing methods unable to handle degradation and the lower subplot showing degradation-handling methods. The red/blue/green indicates the best, runner-up and third best. **Rank** is the mean ranking across all metrics.

| Method | TNO Dataset | | | | | | FMB Dataset | | | | | | Rank↓ |
|---|---|---|---|---|---|---|---|---|---|---|---|---|---|
| | EI↑ | SF↑ | EN↑ | AG↑ | SD↑ | TE↓ | EI↑ | SF↑ | EN↑ | AG↑ | SD↑ | TE↓ | |
| **U2Fusion** (TPAMI'22) | 36.194 | 7.799 | 6.292 | 3.245 | 23.687 | 6.862 | 36.109 | 10.196 | 6.590 | 3.221 | 28.391 | 6.605 | 6.4 |
| **DeFusion** (ECCV'22) | 28.289 | 5.775 | 6.461 | 2.365 | 29.091 | 6.693 | 26.921 | 7.777 | 6.338 | 2.300 | 25.656 | 6.857 | 7.8 |
| **DDFM** (ICCV'23) | 37.096 | 8.665 | 6.871 | 3.421 | 35.093 | 6.283 | 27.841 | 8.542 | 6.597 | 2.470 | 29.014 | 6.598 | 5.1 |
| **EMMA** (CVPR'24) | 51.375 | 11.375 | 7.149 | 4.694 | 45.575 | 6.004 | 46.795 | 14.383 | 6.747 | 4.201 | 34.037 | 6.448 | 1.8 |
| **TC-MoA** (CVPR'24) | 41.128 | 9.250 | 6.777 | 3.613 | 34.397 | 6.377 | 44.411 | 14.045 | 6.756 | 3.974 | 32.412 | 6.439 | 3.5 |
| **CCF** (NeurIPS'24) | 36.462 | 9.448 | 6.804 | 3.448 | 34.708 | 6.350 | 27.560 | 9.029 | 6.452 | 2.467 | 26.145 | 6.743 | 6.0 |
| **DCEvo** (CVPR'25) | 41.786 | 10.038 | 6.767 | 3.752 | 36.215 | 6.387 | 42.498 | 13.715 | 6.697 | 3.773 | 34.042 | 6.498 | 3.8 |
| **PDFuse-B** (Ours) | 52.487 | 12.116 | 7.025 | 4.787 | 40.033 | 6.129 | 47.835 | 14.440 | 6.713 | 4.242 | 34.401 | 6.482 | 1.6 |
| **DDBF** (CVPR'24) | 52.946 | 14.652 | 6.394 | 4.965 | 35.384 | 6.760 | 44.657 | 15.434 | 5.627 | 3.842 | 20.175 | 7.568 | 4.2 |
| **Text-IF** (CVPR'24) | 48.113 | 11.547 | 7.051 | 4.480 | 43.358 | 6.103 | 47.461 | 15.063 | 6.660 | 4.235 | 31.488 | 6.535 | 3.2 |
| **MRFS** (CVPR'24) | 39.513 | 9.223 | 7.174 | 3.549 | 45.977 | 5.979 | 36.958 | 11.831 | 6.740 | 3.257 | 34.313 | 6.456 | 3.4 |
| **TIMFusion** (TPAMI'24) | 41.144 | 9.749 | 7.095 | 3.831 | 44.298 | 6.059 | 41.952 | 13.396 | 6.571 | 3.743 | 29.232 | 6.624 | 3.7 |
| **Text-DiFuse** (NeurIPS'24) | 32.669 | 7.230 | 6.991 | 2.738 | 47.074 | 6.162 | 36.024 | 11.676 | 6.920 | 3.162 | 37.415 | 6.275 | 4.1 |
| **PDFuse-E** (Ours) | 79.869 | 18.300 | 7.079 | 7.499 | 40.303 | 6.075 | 72.684 | 21.307 | 6.549 | 6.557 | 32.504 | 6.646 | 2.5 |

We randomly selected 200 images from the SICE dataset [2] for further evaluation. The qualitative results are presented in Fig. 18, while the quantitative findings are summarized in Table 12. Our fusion method outperforms current state-of-the-art approaches in overall exposure balance, color fidelity, and detail preservation, and also demonstrates superior performance across the aggregate evaluation metrics. To enhance the comparison comprehensiveness, we additionally introduce MEF-SSIM [32] for evaluation. MEF-SSIM is a structural similarity evaluation metric specifically designed for MEF, which has been widely adopted and recognized in the MEF field. This metric performs perception-driven quality assessment by measuring the consistency between the fusion result and the ideal structures of multi-source exposure images. The quantitative experimental results are shown in Table 13.

Table 12: Quantitative comparison of multi-exposure image fusion methods on the SICE dataset [2]. The red/blue/green denotes first, second and third best. **Rank** is the mean ranking across all metrics.

| Methods | EI↑ | SF↑ | EN↑ | AG↑ | SD↑ | TE↓ | SCD↑ | $Q_{cb}$↑ | Rank↓ |
|---|---|---|---|---|---|---|---|---|---|
| **U2Fusion** | 77.570 | 21.379 | 7.119 | 7.341 | 50.364 | 5.461 | 0.799 | 0.475 | 4.5 |
| **DeFusion** | 73.184 | 20.298 | 7.102 | 6.884 | 50.023 | 5.478 | 0.030 | 0.426 | 6.4 |
| **HSDS-MEF** | 68.284 | 24.476 | 7.262 | 8.278 | 51.805 | 5.318 | 0.725 | 0.477 | 3.5 |
| **TC-MoA** | 69.101 | 17.624 | 7.154 | 6.436 | 45.080 | 5.426 | 0.423 | 0.485 | 5.8 |
| **CCF** | 74.602 | 24.061 | 7.328 | 7.478 | 56.988 | 5.252 | 1.186 | 0.461 | 3.1 |
| **SAMT-MEF** | 76.682 | 24.253 | 7.205 | 7.811 | 47.788 | 5.375 | 0.612 | 0.486 | 3.6 |
| **EAT** | 61.909 | 19.253 | 7.063 | 6.245 | 43.476 | 5.517 | 0.345 | 0.426 | 7.6 |
| **PDFuse-B** | 90.641 | 26.568 | 7.461 | 8.828 | 58.199 | 5.119 | 0.877 | 0.482 | 1.4 |

### E.3 More evaluations for multi-focus image fusion

Table 13: Comparison with advanced methods based on the MEF-SSIM metric on the SICE dataset.

| Methods | U2Fusion | DeFusion | HSDS-MEF | TC-MoA | CCF | SAMT | EAT | PDFuse |
|---------|----------|----------|----------|--------|-----|------|-----|--------|
| **MEF-SSIM** | 0.8030 | 0.8297 | 0.8164 | 0.8838 | 0.7963 | 0.8617 | 0.8574 | **0.8856** |

We further present a comprehensive evaluation of multi-focus image fusion performance on the widely-used Lytro dataset [27], with qualitative results in Fig. 14 and quantitative results in Table 14. Our method not only demonstrates superior performance over the peer CCF approach but also outperforms all other competing algorithms, achieving the highest average ranking across every quantitative metric.

Table 14: Quantitative comparison of multi-focus image fusion methods on the Lytro dataset. The **red**/**blue**/**green** denotes first, second and third best. **Rank** is the mean ranking across all metrics.

| Methods | EI↑ | SF↑ | EN↑ | AG↑ | SD↑ | TE↓ | SCD↑ | $Q_{cb}$↑ | Rank↓ |
|---------|-----|-----|-----|-----|-----|-----|------|-----------|-------|
| U2Fusion | 62.670 | 14.890 | 7.304 | 5.589 | 51.853 | 7.673 | 0.373 | 0.648 | 5.8 |
| DeFusion | 48.944 | 11.117 | 7.468 | 4.421 | 54.483 | 7.509 | 0.330 | 0.597 | 6.6 |
| ZMFF | 74.427 | 18.921 | 7.525 | 6.755 | 56.800 | 7.452 | 0.467 | 0.740 | 4.3 |
| DB-MFIF | 75.006 | 19.641 | 7.537 | 6.895 | 57.654 | 7.439 | 0.576 | 0.778 | 3.1 |
| TC-MoA | 75.280 | 19.443 | 7.548 | 6.895 | 58.340 | 7.428 | 0.726 | 0.733 | 2.8 |
| CCF | 53.997 | 13.526 | 7.592 | 4.767 | 62.389 | 7.384 | 1.338 | 0.598 | 3.8 |
| PDFuse-B | 81.497 | 19.854 | 7.613 | 7.266 | 61.799 | 7.364 | 0.818 | 0.616 | 1.8 |

# F Extended application

## F.1 Evaluations for medical multimodal image fusion

To further validate the generality of our approach, we performed extended evaluations in the context of medical multimodal image fusion. Specifically, we employed the CT–MRI fusion subset of the Harvard Medical School dataset as a representative benchmark. Our method was compared against both state-of-the-art multimodal fusion algorithms and widely used general-purpose image fusion techniques.

Table 15: Quantitative comparison of medical image fusion methods on the Havard medicine dataset. The **red**/**blue**/**green** denotes first, second and third best. **Rank** is the mean ranking across all metrics.

| Methods | EI↑ | SF↑ | EN↑ | AG↑ | SD↑ | TE↓ | SCD↑ | $Q_{cb}$↑ | Rank↓ |
|---------|-----|-----|-----|-----|-----|-----|------|-----------|-------|
| U2Fusion | 50.292 | 19.931 | 4.335 | 4.762 | 73.959 | 1.443 | 1.015 | 0.550 | 5.1 |
| DeFusion | 56.435 | 19.537 | 4.497 | 5.321 | 52.312 | 1.282 | 0.542 | 0.284 | 5.6 |
| DDFM | 46.170 | 19.226 | 4.038 | 4.465 | 63.580 | 1.740 | 0.995 | 0.574 | 6.0 |
| EMMA | 67.381 | 23.162 | 4.995 | 6.449 | 82.784 | 0.783 | 1.409 | 0.494 | 2.6 |
| TC-MoA | 57.787 | 25.956 | 4.480 | 5.668 | 74.192 | 1.298 | 1.242 | 0.291 | 4.3 |
| CCF | 64.751 | 22.358 | 5.187 | 6.015 | 87.586 | 0.592 | 1.474 | 0.385 | 2.6 |
| PDFuse-B | 76.292 | 32.205 | 4.540 | 7.279 | 90.330 | 1.238 | 1.516 | 0.503 | 1.7 |

Qualitative comparisons are presented in Fig. 20, and quantitative results are summarized in Table 15. The results demonstrate that the proposed approach not only preserves critical anatomical structures with high fidelity but also accurately reflects regional variations in metabolic activity.

## F.2 Evaluations for complex weather degradation scenarios

We extracted challenging complex-weather degradation scenes from the M3FD infrared–visible image fusion dataset, which was captured using a dual-lens optical camera and a dual-lens infrared sensor. This dataset contains adverse environmental conditions such as haze combined with low light, low light accompanied by headlight glare, and haze mixed with noise. To evaluate our proposed PDFuse-E method, we compared it with two cascaded schemes. In the first scheme, the visible image was pre-enhanced using the DA-RCOT restoration method [43] before being fused with DCEvo. In the second scheme, the visible image was pre-enhanced using the OneRestore method [10] and then fused with DCEvo. The quantitative comparison results are presented in Table 16.

Table 16: Quantitative comparison on challenging complex-weather degradation scenes from the M3FD dataset.

| Methods | EI↑ | SF↑ | AG↑ | EN↑ | SD↑ | TE↓ | SCD↑ | $Q_{cb}$↑ |
|---------|-----|-----|-----|-----|-----|-----|------|-----------|
| **DA-RCOT + DCEVo** | 67.56 | 17.94 | 6.31 | 6.78 | 32.71 | 6.89 | 1.57 | 0.51 |
| **OneRestore + DCEvo** | 105.45 | 27.90 | 9.86 | 7.14 | 42.20 | 6.53 | 0.96 | 0.45 |
| **PDFuse-E** | 96.07 | 24.54 | 9.08 | 6.88 | 35.14 | 6.79 | 1.35 | 0.49 |

From the table, we observe that the OneRestore + DCEvo method achieves the best performance on several non-reference metrics such as EI, SF, and AG, but yields relatively low scores on full-reference metrics like SCD and Qcb. In contrast, the DA-RCOT + DCEvo method performs best on SCD and Qcb, yet falls behind on non-reference metrics such as EI and SF. While our method does not outperform the others on any single metric, it achieves a more balanced overall performance. For example, our scores on non-reference metrics like EI and AG are very close to those of OneRestore + DCEvo and significantly higher than those of DA-RCOT + DCEvo. At the same time, our full-reference scores on SCD and Qcb remain relatively high.

From a qualitative perspective as illustrated in Fig. 10, in low-light and extreme haze-degraded scenes, both OneRestore and DA-RCOT struggle to restore visibility in the hazy regions of the visible image, resulting in fused outputs that retain heavy haze and hinder the integration of infrared information. While our PDFuse-E method also cannot directly restore haze in the visible image, the interference operator set adjusts the global exposure structure, effectively suppressing haze brightness and thus reducing its fusion weight. This enables the fused output to preserve rich infrared texture information in those regions.

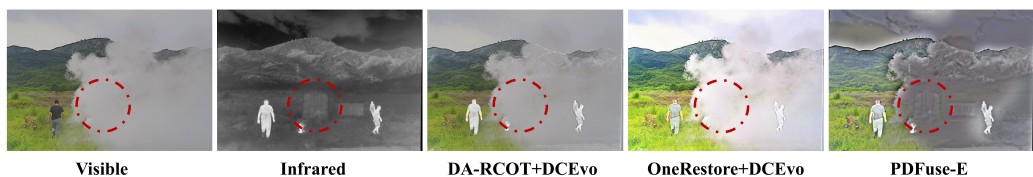

| Visible | Infrared | DA-RCOT+DCEvo | OneRestore+DCEvo | PDFuse-E |

Figure 10: Qualitative comparison under complex weather degradations.

# G  Limitation

Although our method is the first training-free, general, and robust image fusion framework based on pretrained latent diffusion models and achieves good performance across multiple tasks, it also shares a common limitation with most diffusion model methods, namely, relatively low runtime efficiency. In Table 17, we provide the memory usage and runtime efficiency of various methods. Regarding runtime efficiency, several diffusion model methods perform poorly. Although our method runs faster compared to similar approaches like CCF and DDFM, there remains significant room for improvement compared to other architectural methods.

Table 17: Statistical results of parameters and runtime.

| Methods | U2Fusion | DeFusion | TC-MoA | EMMA | DCEvo | DDFM | CCF | PDFuse-B | PDFuse-E |
|---|---|---|---|---|---|---|---|---|---|
| Memory usage/M | 2.51 | 30.04 | 1299.23 | 5.79 | 7.65 | 2108.82 | 2108.82 | 2033.68 | 2033.68 |
| Runtime/Second | 1.50 | 0.92 | 1.14 | 1.12 | 0.74 | 57.69 | 96.94 | 18.43 | 18.81 |

# H  Broader impacts

This paper proposes a training-free, generalizable, and robust image fusion method leveraging a pretrained latent diffusion model to generate high-quality fused images that serve both visual perception and machine decision-making. Specifically, the study aims to advance the exploration and application of latent diffusion models in the field of image fusion, addressing two major challenges currently faced by image fusion tasks: generalization and robustness. It is foreseeable that this work will have a positive impact across various domains, particularly facilitating the advancement of information fusion technologies toward practical applications such as autonomous driving and intelligent medical diagnosis. The proposed approach poses minimal risks and is unlikely to have any significant negative effects.

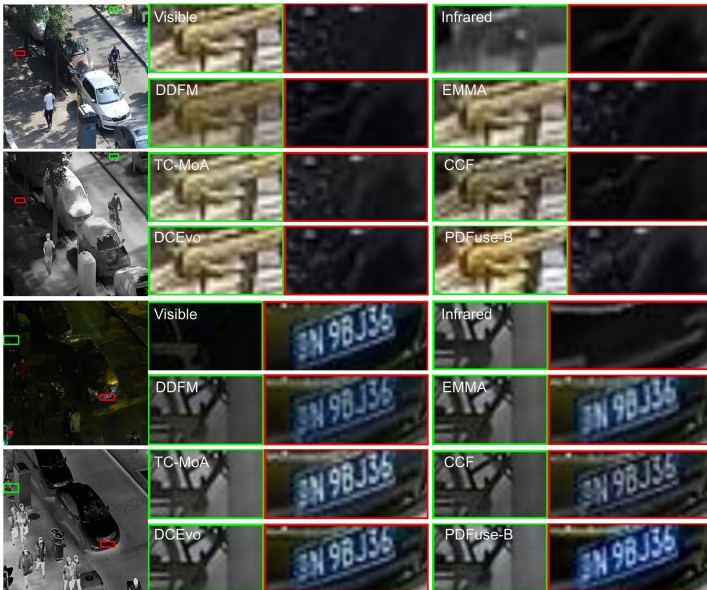

Figure 11: Center zoomed-in views of qualitative infrared–visible image fusion results.

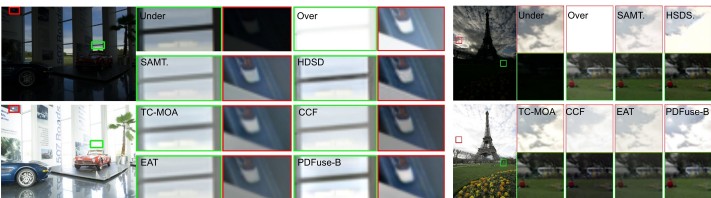

Figure 12: Center zoomed-in views of qualitative multi-exposure image fusion results.

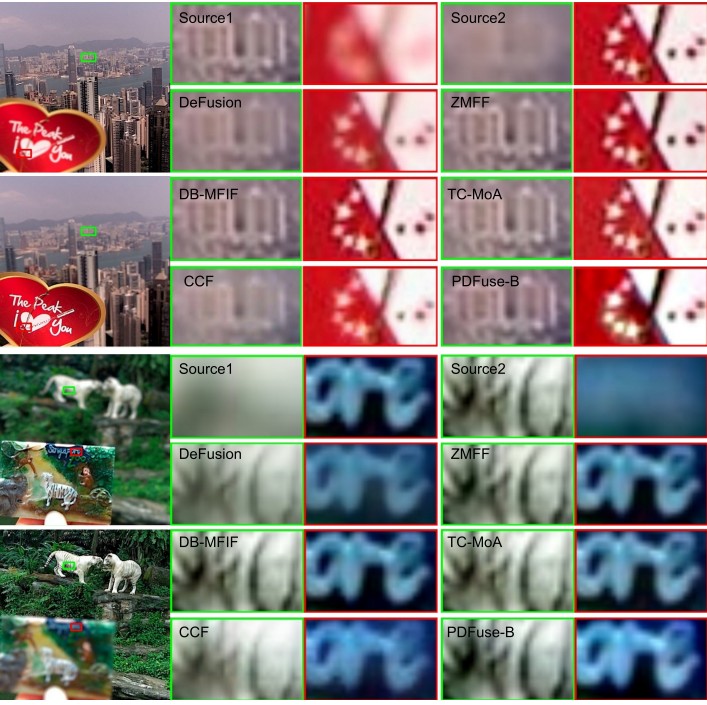

Figure 13: Center zoomed-in views of qualitative multi-focus image fusion results.

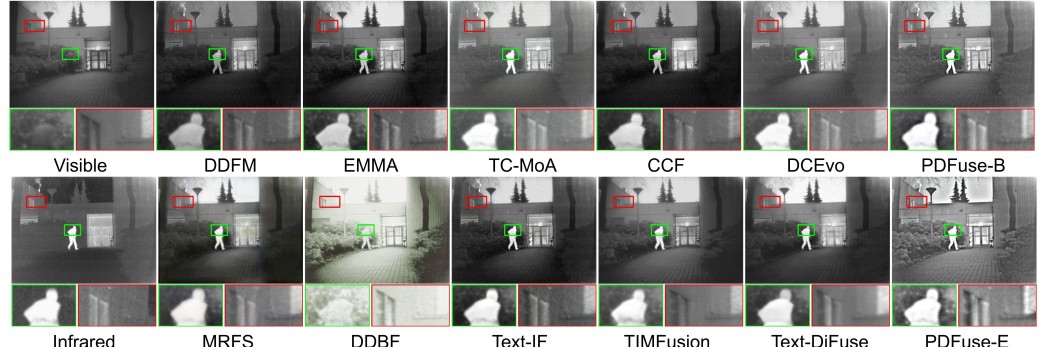

Figure 14: Qualitative comparison of PDFuse-B, PDFuse-E, and state-of-the-art methods on the TNO dataset.

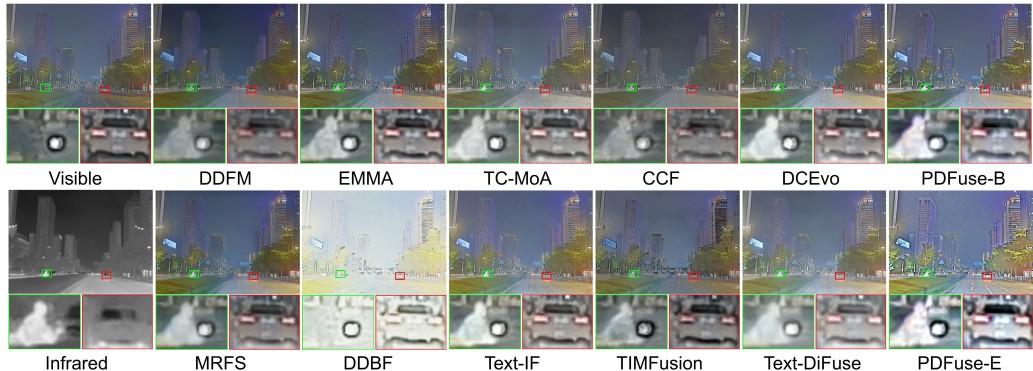

Figure 15: Qualitative comparison of PDFuse-B, PDFuse-E, and state-of-the-art methods on the FMB dataset.

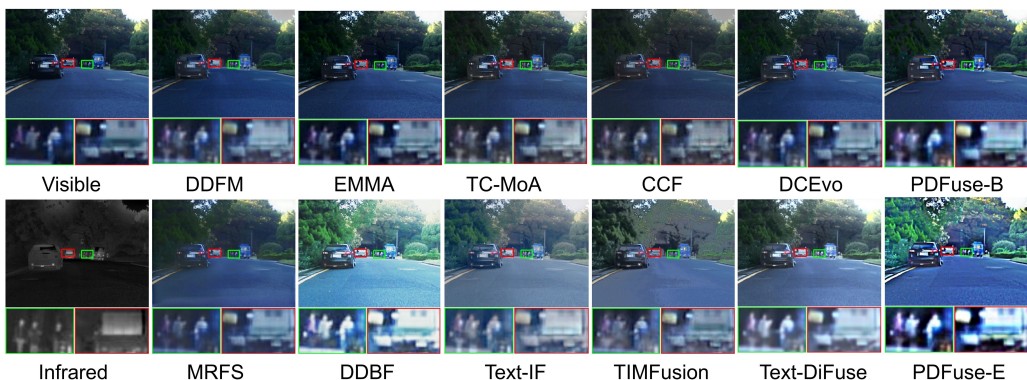

Figure 16: Qualitative comparison of PDFuse-B, PDFuse-E, and state-of-the-art methods on the MSRS dataset.

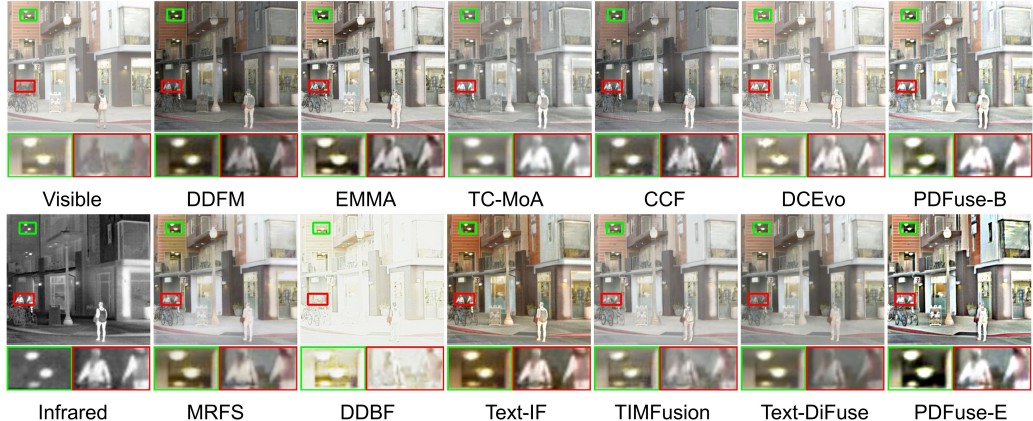

Figure 17: Qualitative comparison of PDFuse-B, PDFuse-E, and state-of-the-art methods on the RoadScene dataset.

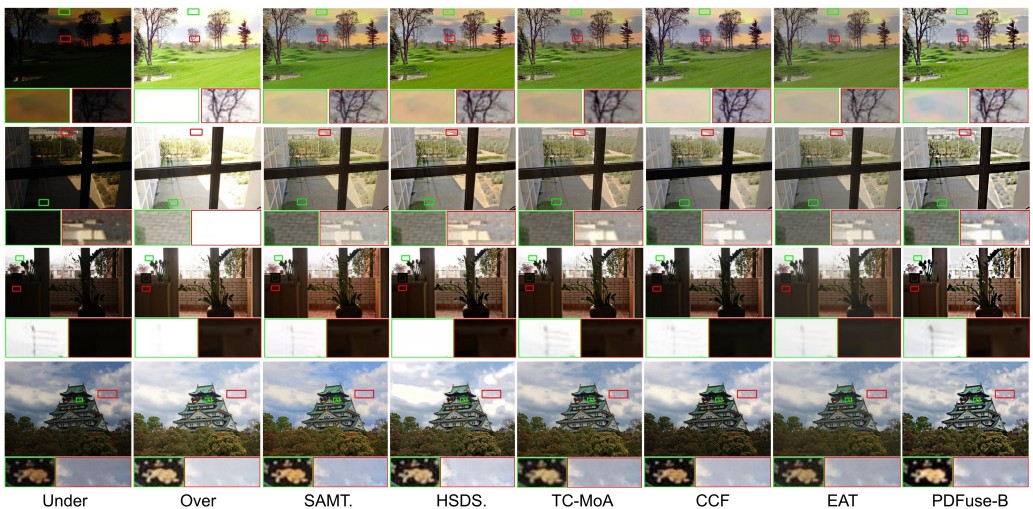

Figure 18: Qualitative comparison of PDFuse-B and state-of-the-art methods on the SICE dataset.

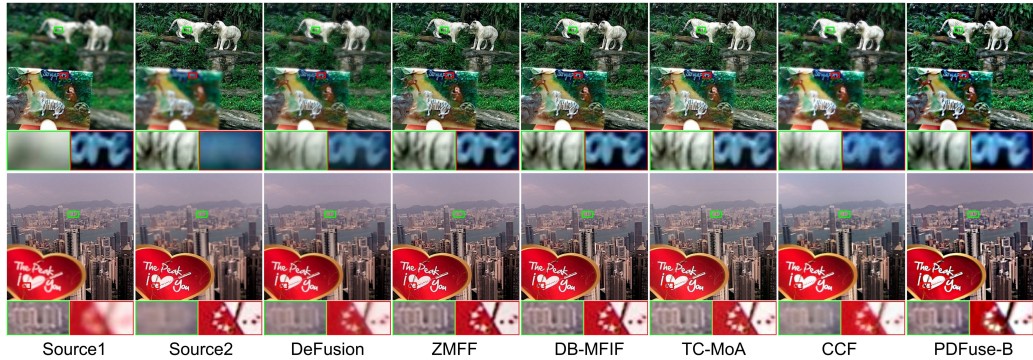

Figure 19: Qualitative comparison of PDFuse-B and state-of-the-art methods on the Lytro dataset.

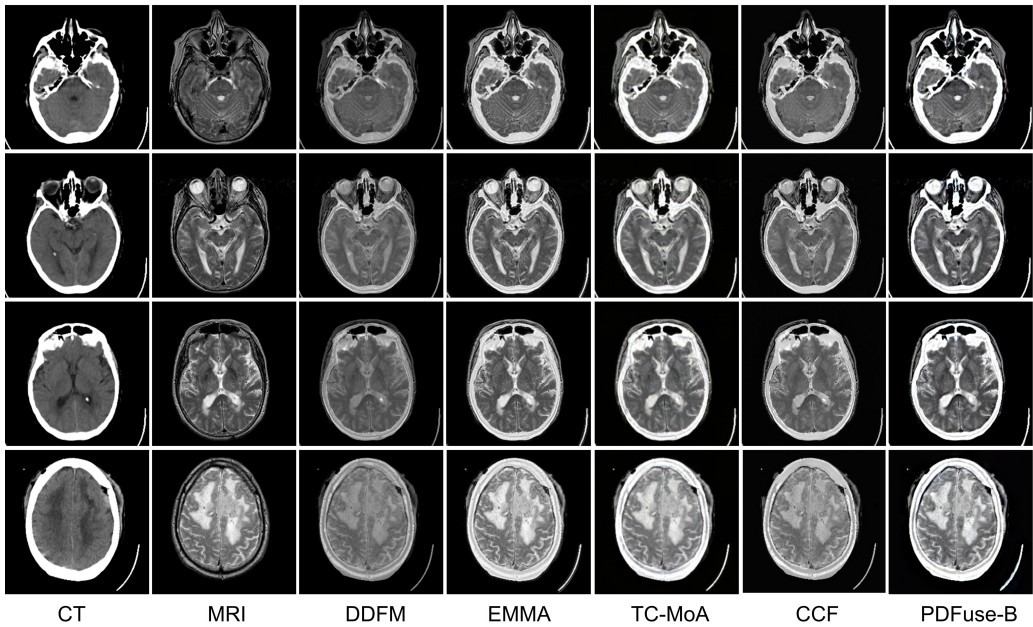

| CT | MRI | DDFM | EMMA | TC-MoA | CCF | PDFuse-B |

Figure 20: Qualitative comparison of PDFuse-B and state-of-the-art methods on the Havard medicine dataset.

