# OpenReview forum: "Projection-Manifold Regularized Latent Diffusion for Robust General Image Fusion"
_NeurIPS.cc/2025/Conference — NeurIPS 2025 poster_

### Official Review · Reviewer_2sqp · 2025-06-11

**Clarity:** 2
**Significance:** 3
**Originality:** 3
**Rating:** 2
**Confidence:** 5

**Summary:**

This paper proposes a training-free and unified framework, named PDFusion, which incorporates the diffusion posterior sampling process into a pre-trained latent diffusion model. It introduces MICP and LMG to guide the generation process, aiming to improve fidelity and quality. Additionally, the framework claims the ability to handle degraded inputs inherently. Extensive experiments across various fusion scenarios and downstream tasks demonstrate the method's robustness.

**Questions:**

- Empirically, the pre-trained VAE is unable to generate high-quality visible and infrared fused images, as it has never been exposed to such data during training. It remains unclear how the model is guided in the latent space and how it decodes these representations to synthesize high-fidelity fusion images. Additionally, Appendix A.3 is difficult to follow: the algorithm references Equations (10)–(15), but these equations are either missing or poorly integrated, resulting in a presentation that appears chaotic and incomprehensible.
- The performance of PDFuse-E appears questionable, both in terms of underlying principles and the reported metric values.

**Ethical Concerns:**

["NO or VERY MINOR ethics concerns only"]

**Final Justification:**

- First, my main question regarding the rationale behind the unusually large performance gains of PDFuse-E, which reportedly doubles or even triples the performance of prior SOTA methods, remains unanswered. Such substantial improvements appear revolutionary, yet they are achieved solely with a training-free model. This raises serious concerns about the fairness and reliability of the metrics presented in the paper and significantly affects my confidence in the reported results.

- Second, although LDMs are generally faster than DDPMs due to latent-space denoising, the proposed method employs a decoder at every denoising step to reconstruct latent images—an architectural choice that likely negates the typical speed advantage of LDMs. Moreover, the comparison is made only against current DDPM-based approaches, which is an unfair baseline and risks overstating the claimed efficiency gains. In addition, the authors’ assertion that “our runtime remains nearly constant as the number of input images increases, while that of others grows roughly linearly” is unconvincing and counterintuitive, further undermining my confidence in the authenticity of the reported results.

- Third, the degradation operator is applied to the decoded $x$; however, this approach is not grounded in the core principles of LDMs.

Given that these core issues remain unresolved, I will retain my original score.

**Limitations:**

Yes

**Paper Formatting Concerns:**

There is no major formatting issues in this paper.

**Quality:**

3

**Strengths And Weaknesses:**

### Strengths

- The motivation is novel. PDFusion leverages a pre-trained latent diffusion model to guide the fusion process in the latent space.
- The use of a canonical interference operator set allows PDFusion to handle various types of degradation and produce higher-quality fusion results.

### Weaknesses

- PDFusion is based on Stable Diffusion but is guided at the pixel level. In other words, the fusion occurs at the pixel level and is subsequently encoded into the latent space. This approach undermines the advantage of latent-space fusion, making it fundamentally similar to conventional DDPM approaches.
- Table 1 reports an unusually large performance gain for PDFuse-E in metrics such as EI, SF, and AG. It is difficult to understand how degraded inputs, which have lost information, can produce outputs of higher quality than those generated from clean inputs. This discrepancy raises concerns regarding the credibility and reliability of the reported results.
- While the method is described as training-free, it remains unclear whether the de-degraded outputs are guided using clean inputs. If so, this undermines the validity of the "training-free" claim. If not, it is unclear how the method ensures clean fusion outputs under pixel-level constraints with degraded inputs.
- The paper does not clearly explain how the datasets $\mathcal{D}^s$ and $\mathcal{D}^d$ are constructed or obtained.
- Every generation step includes a decoding operation. Given this, it is difficult to understand how PDFusion achieves faster runtime than DDFM and CCF, as reported in Table S6.

>  If the authors can clarify my concerns, I would be inclined to significantly raise my score.

---

> ### Author Rebuttal · Authors · 2025-07-31
>
> **Q1**: Explanation of $\mathcal{D}^s$ and $\mathcal{D}^d$.
>
> **Reply**: $\mathcal{D}^s$ and $\mathcal{D}^d$ are the two projection stages of the MICP regularization constraint, not additional datasets. Specifically, to mitigate the gradient conflict caused by simultaneously optimizing intensity loss and gradient loss directly on the fused image and to enhance the efficiency of the regularization, we divided the regularized projection of the MICP mechanism into two stages. In the first stage, we focused on optimizing intensity loss to obtain the coarse structural component $\mathcal{D}^s(\hat{\mathbf{z}}_0^t)$ as in Eq. (8), abbreviated as $\mathcal{D}^s$. In the second stage, we treated texture as an additional layer optimization to ensure the most detailed texture while maintaining the overall intensity structure unchanged, generating a fine detail component $\mathcal{D}^d(\hat{\mathbf{z}}_0^t)$, abbreviated as $\mathcal{D}^d$ as in Eq. (10). The two projection components were combined to produce the final projection result.
>
> **Q2**: Explanation on VAE Fidelity.
>
> **Reply**:  Actually, VAE does exhibit significant information bias and loss when applied to high-quality visible images with very dense and regular textures. However, the loss introduced by VAE is typically concentrated in its high-frequency components (TVT, ICCV 2025). We performe a 2D FFT on both the original image and the VAE-encoded and decoded reconstructed image, generating a radial frequency grid followed by frequency band partitioning, which is used to study the distribution of MSE losses from VAE encoding and decoding across different normalized frequency bands on the LLVIP dataset, as shown below:
>
>
> |Frequency Range (cycles/pixel)|0.00-0.025|0.025-0.050|0.050-0.100|0.100-0.250|0.250-0.500|
> |-|-|-|-|-|-|
> | MSE|0.000000|0.000001|0.000003|0.000062|0.001003|
>
> It is clear that VAE's loss and reconstruction bias are concentrated in the high frequencies. To address this challenge, we adopted a high-frequency residual injection strategy. Specifically, in our MICP mechanism, the regularized projection captures the coarse structural component $\mathcal{D}^s(\hat{\mathbf{z}}_0^t)$ and the fine detail component $\mathcal{D}^d(\hat{\mathbf{z}}_0^t)$ from the multi-source image set {$i^m, i^n$}. Then, a high-frequency filter $\mathcal{G}(\cdot)$ is applied to remove the high-frequency components from the projection result as in Eq. (11). In other words, the high-frequency information is kept as $R$ rather than encoded into the latent space. Moreover, $R$ is iteratively refined via projection regularization of $\hat{\mathbf{z}}_0^{t-1}$ and finally injected as a residual after the process completes. (see Appendix A.3).
>
> Additionally, we further conduct an ablation study on the high-frequency filtering and residual structure. The MICP mechanism projection result is directly encoded by VAE into the latent space (without pre-reserving high-frequency information), and the quantitative comparison is shown in the table below:
>
> |Method|EI↑|SF↑|EN↑|AG↑|SD↑|TE↓|SCD↑|Q_cb↑|
> |-|-|-|-|-|-|-|-|-|
> |wo/(filtering & residual) |39.391|8.070|7.359|3.371|45.077|7.528|0.913|0.418|
> |PDFuse-B|48.519|11.028|7.427|4.221|47.931|7.457|0.981|0.437|
>
> Based on the ablation results, the model variant lacking high-frequency filtering and residual components exhibits a significant loss in edge and texture details. These results demonstrate the effectiveness of our adopted high-frequency residual injection strategy.
>
> **Q3**: Explanation of the algorithm process in Appendix A.3.
>
> **Reply**: The equations referenced in the algorithm in Appendix A.3 correspond to those presented in the main text, rather than any missing ones. We apologize for the misunderstanding. We added the symbol $†$ to distinguish these equations from those in the appendix and provided an explanation after the algorithm.
>
> **Algorithm process:** After each step of diffusion, the intermediate variable $\hat{\mathbf{z}}_0^t$ is obtained and then constrained through the alternating optimization regularization mechanism. If the MICP mechanism is adopted, the first-stage projection is performed using Eq. (8) in the main text to obtain $\mathcal{D}^s(\hat{\mathbf{z}}_0^t)$, followed by the second-stage projection using Eq. (10) to obtain $\mathcal{D}^d(\hat{\mathbf{z}}_0^t)$. After updating the high-frequency residual $R$, the projection result is weighted and encoded into the latent space using Eq. (11), with the gradient of LMG mechanism set to 0. If the LMG mechanism is adopted, Eq. (12) is used to establish the consistency gradient between pixel space and latent space to guide $\hat{\mathbf{z}}_0^t$, while the MICP mechanism performs the identity projection of $\hat{\mathbf{z}}_0^t$. The update of $\mathbf{z}_t$ is completed by applying the regularization constraint with Eq. (6). The PDFuse-E process follows the same steps as PDFuse-B, with the distinction that its MICP and LMG mechanisms in Eq. (14) ~ Eq. (16) integrate the interference operator set $\mathbf{\Pi}$.
>
> In the camera ready, we will merge the main text and the appendix, unify the numbering of the equations, and include more detailed explanations of the algorithm process to enhance the clarity of the algorithm description.
>
> **Q4**: Time Efficiency Analysis of PDFuse, CCF, and DDFM.
>
> **Reply**: In all experiments, we adhered to the default settings from the DDFM and CCF papers, with diffusion steps set to 100 and 200, respectively, while our method, PDFuse, was configured with 50 steps. Despite using fewer diffusion steps than DDFM and CCF, PDFuse outperforms both methods across all tasks. Furthermore, we compared PDFuse, CCF, and DDFM with a uniform diffusion step count of 100, and the resulting running time comparisons are shown in the table below:
>
> |Method|Runtime (100 steps) / Second|
> |-|-|
> |DDFM|55.29|
> |CCF|50.51|
> |PDFuse|**32.63**|
>
> Our method is still faster than DDFM and CCF, which is due to the higher efficiency of latent space diffusion compared to pixel space diffusion.
>
> **Q5**: Explanation of the principles and performance of PDFuse-E.
>
> **Reply**:  First, we would like to clarify that our approach does not require any training, which achieves image fusion through an alternating regularized existing latent diffusion process. Secondly, our method does not rely on any clean input to guide the degradation removal. Instead, it addresses the degradation by integrating the interference operator set. Next, we will provide a detailed clarification and illustration of this approach.
>
> Actually, when multi-source images $\{i^m, i^n\}$ suffer from degradations (such as Gaussian blur), the loss (such as the maximum intensity loss $min_{\hat{\mathbf{z}}_0^t}\left\| \mathcal{D}(\hat{\mathbf{z}}_0^t) - \max(i^m, i^n) \right\|_2^2$) will also cause the final result $\mathcal{D}({\mathbf{z}}_0)$ to also exhibit degraded. In this context, our PDFuse-E integrates the interference operator set $\mathbf{\Pi}$ into the regularization constraints (such as explicitly modeling the Gaussian blur operator $\mathbf{\Pi}_g$), and the maximum intensity loss is transformed into:
>
> $min_{\hat{\mathbf{z}}_0^t}\left\| \mathbf{\Pi}_g \mathcal{D}(\hat{\mathbf{z}}_0^t) - \max(i^m, i^n) \right\|_2^2$.
>
> In this case, the maximum intensity loss constraint can guide the LDMs to solve this inverse problem, generating a fused image that does not contain Gaussian blur. This technique, which explicitly models the interference operator set and solves it using diffusion priors, has achieved remarkable results in the unsupervised image restoration field. (PSLD, NeurIPS 2023; LLIEDiff, AAAI 2025). We are the first to extend the aforementioned concept and apply it to address degradations in the multi-source image fusion process. Through careful design, we integrate the interference operator into various components of the MICP and LMG mechanisms, enabling our method to effectively handle challenging degradation scenarios such as low light, exposure issues, blur, and thermal diffusion. This results in significant improvements in metrics such as EI, SF, and AG.
>
> **Q6**: The advantages of our approach compared to the existing DDPM method.
>
> **Reply**: Compared to DDPM, LDMs offer a stronger generative prior, but their regularization constraints are more complex. The main contribution of our method is the design of an adaptive multi-source image regularization mechanism based on the generative prior of LDMs. To the best of our knowledge, this is the first **training-free**, **general**, and **robust** image fusion method that leverages pre-trained LDMs. Unlike existing CCF and DDFM methods, our approach is distinct in the following ways: **(1)** Our method leverages the stronger generative priors of LDMs. **(2)** We designed a more efficient MICP projection regularization mechanism, which decomposes the projection of $\mathbf{z}_0^t$ into a multi-layer optimization of the coarse structure and fine detail layers, effectively alleviating gradient conflicts when optimizing multiple objective losses within the same layer. **(3)** The LMG mechanism we designed simultaneously applies regularization guidance in both pixel and latent spaces with multi-source images, effectively accommodating the many-to-one mapping of the VAE decoder in LDMs. Since the VAE decoder exhibits a many-to-one mapping, applying multi-source image guidance solely in pixel space results in multiple optimization directions for the latent space variable $\mathbf{z}_0^t$. Introducing latent space regularization ensures that $\mathbf{z}_0^t$ is optimized in a specific direction. **(4)** We designed a flexible alternating optimization strategy that balances multi-source information fidelity with the preservation of latent diffusion flow. **(5)** The regularization mechanism can naturally integrate the interference operator set, enabling it to handle various degradations in the multi-modal image fusion process, with strong robustness.

---

> > ### Comment · Reviewer_2sqp · 2025-08-04
> > **Thank you for your response**
> >
> > Thank you for your response. Some of my concerns have been partially addressed. The authors have not yet provided a satisfactory explanation for the substantial improvements in the EI, SF, and AG metrics achieved by PDFuse-E, which reportedly doubles or even triples the performance compared to current SOTA methods. What is the underlying rationale or mechanism that justifies such significant gains? In addition, the design and operational details of the proposed MICP mechanism remain unclear and require further clarification.

---

> > > ### Author Response · Authors · 2025-08-05
> > > **Clarification of EI, SF, and AG enhancements and MICP mechanism details.**
> > >
> > > We appreciate your feedback and offer the following clarification.
> > >
> > > **Q1**：Analysis of the improvements in the EI, SF, and AG indicators.
> > >
> > > **Reply**：EI, SF, and AG are no-reference metrics used to evaluate fusion quality from structural, frequency, and texture detail perspectives, respectively, and they are closely correlated with the overall scene quality of the fused images. Our test set comprises datasets with diverse and complex degradation types, such as low-light, blur, and thermal diffusion. If these degradations are not effectively addressed during fusion, they can affect the characteristics (structural, frequency, and texture) of the fused results, thereby degrading the performance in metrics like EI, SF, and AG. For example, low‐light conditions diminish gradient magnitudes, thereby reducing the AG metric; blur suppresses high‐frequency components, leading to a lower SF value.
> > >
> > > We carefully selected state-of-the-art methods capable of handling degradations, including Text-DiFuse, Text-IF, and TIMFusion. However, these methods rely on supervised training on degraded datasets and cannot explicitly model the distributions of different degradation types. As a result, they struggle to handle all degradation types in complex scenarios, leading to the retention of certain degradations and impairing the structural, frequency, and texture details of the fused images, ultimately resulting in lower EI, SF, and AG scores. In contrast, our method explicitly models various degradation types and integrates them into the MICP and LMG mechanisms. By leveraging powerful diffusion priors from LDMs to solve the degradation inverse problem, it effectively addresses diverse degradations and generates rich structural and texture details as in Fig. (4). Consequently, PDFuse-E achieves significant improvements in the texture-related metrics EI, SF, and AG.
> > >
> > >
> > > **Q2**：Detailed of the MICP Mechanism.
> > >
> > >  **Reply**：We provide additional details in Appendix A.1 of the supplementary material on the MICP mechanism, and here offer further explanation.
> > >
> > >   The MICP mechanism adopts the mainstream loss in image fusion, comprising “pixel intensity loss” and “gradient loss” (Appendix Eq. (1)). The multi-objective optimization of these two losses renders the overall process non-convex. We decompose the loss into two stages: one optimizing the pixel intensity loss for the coarse structural component $\mathcal{D}^s(\hat{\mathbf{z}}_0^t)$, simplified as $\mathcal{D}^s$, and the other optimizing the gradient loss for the fine detail component $\mathcal{D}^d(\hat{\mathbf{z}}_0^t)$, simplified as $\mathcal{D}^d$, with each loss optimized on an independent component. The final projected result is obtained by summing these two components. Below, we detail each stage.
> > >
> > > **Details of the first stage** are in Stage 1 of Appendix A.1. To reach the optimum, we differentiate the pixel intensity loss, yielding Appendix Eq. (2). This equation is then solved via conjugate gradient to obtain the $\mathcal{D}^s$. **Details of the second stage** are in Stage 2 of Appendix A.1. An additional projection layer, the $\mathcal{D}^d$, is optimized such that the total projection $\mathcal{D}^s + \mathcal{D}^d$ satisfies the maximum pixel intensity and maximum gradient. Therefore, the optimization of $\mathcal{D}^d$ must adhere to the following constraints:
> > >
> > > (1): $\min_{\mathcal{D}^d} \left\| \nabla \left( \mathcal{D}^s + \mathcal{D}^d \right) - \max\left( \nabla i^m, \nabla i^n \right) \right\|_2^2$. This ensures the gradient of the total projection $(\mathcal{D}^s + \mathcal{D}^d)$ aligns with the maximum gradient across source images. This corresponds to the detail-constraint term in Appendix Eq. (3).
> > >
> > > (2): Since the $\mathcal{D}^s$ has already been optimized to satisfy:
> > >
> > > $\min_{\mathcal{D}^s} \left\| \mathcal{D}^s - \max(i^m, i^n) \right\|_2^2$,
> > >
> > > the $\mathcal{D}^d$ only needs to be optimized according to $\min_{\mathcal{D}^d} \left\| \mathcal{D}^d \right\|_2^2$ in order for the sum $(\mathcal{D}^s + \mathcal{D}^d)$ to satisfy the maximum pixel intensity. This corresponds to the regularization term 1 in Appendix Eq. (3).
> > >
> > > To suppress grid-like artifacts in the $\mathcal{D}^d$, an additional regularization term 2 is introduced, forming the overall optimization objective(Appendix Eq. (3)). By deriving and simplifying this objective, we obtain the second-stage equation (Appendix Eq. (6)). Appendix Eq. (7) and Eq. (8) further prove the convergence and global optimality of both projection equations, thereby providing a foundation for integrating the interference operator set into the MICP mechanism. The total loss after incorporating degradation operators is defined in Appendix Eq. (9), and its derivation to the two-stage projection remains consistent with the above process, yielding the final projection equations with the integrated interference operator set (Main text Eq. (14) and Eq. (15)).

---

> > > > ### Comment · Reviewer_2sqp · 2025-08-06
> > > > **Thanks for your response.**
> > > >
> > > > Thank you for your response. However, the authors have not fully addressed my concerns.
> > > >
> > > > First, my main question regarding the rationale behind the unusually large performance gains of PDFuse-E, which reportedly doubles or even triples the performance of prior SOTA methods, remains unanswered. As the method is training-free, such dramatic improvements raise concerns about the reliability and validity of the reported results. This significantly affects my confidence in the fairness of the metric presented in the paper.
> > > >
> > > > Second, although LDMs are typically faster than DDPMs due to latent-space denoising, the proposed method requires a decoder at every denoising step to reconstruct latent images. This architectural design likely offsets the typical speed advantage of LDMs, which has not been adequately discussed. Furthermore,  what is the essential difference between it and the DDPM-based methods?
> > > >
> > > > Third, the MICP mechanism remains unclear. The types of degradation used, such as Gaussian blur, motion blur, and bicubic downsampling, are relatively similar and contradict the paper's claim that PDFuse can effectively handle complex composite degradations. The MICP module appears to apply a Laplacian operator to enhance edges, suggesting that it primarily targets blur. However, this approach is not grounded in the core principles of LDMs.
> > > >
> > > > The issues related to evaluation metrics and degradation settings are also raised by Reviewers 8Yc3 and iFAK.
> > > >
> > > > Given that these core issues remain unresolved, I will retain my original score.

---

> > > > > ### Author Response · Authors · 2025-08-08
> > > > >
> > > > > **4. Large improvements in EI, AG and SF**
> > > > >
> > > > > As explained in our discussion and response, EI, SF, and AG are non-reference metrics that measure structural integrity, frequency content, and local detail, respectively, and are closely related to degradations such as low light and blur. Such degradations significantly reduce the source images' structure, frequency, and detail, causing EI, SF, and AG to drop. If the fusion process fails to remove these degradations, the fused image will also score low on these metrics (as is the case for our method, PDFuse-B, which lacks the interference operator set).
> > > > >
> > > > > PDFuse-E, however, incorporates the interference-operator set into the regularization mechanisms MICP and LMG (Eq. (14)–(17)). By leveraging the generative prior of LDMs to explicitly model the inverse degradation process, PDFuse-E suppresses low-light and blur and reconstructs rich texture details (see Fig. 4). This degradation removal and texture recovery account for the substantial gains observed in EI, SF, and AG.
> > > > >
> > > > > **5. Clarification on differences from DDPM-based methods**
> > > > >
> > > > > As discussed, we have provided five targeted responses addressing distinctions from DDPM-based methods, covering aspects such as the use of LDM priors, the performance of our regularization mechanism, the resolution of the many-to-one mapping in LDM VAEs, and the integration of a degradation-operator set. These collectively demonstrate clear methodological differences, which are further supported by the performance gains observed.
> > > > >
> > > > > **6. Clarification on other reviewers’ concerns**
> > > > >
> > > > > Regarding Reviewer 8Yc3’s concern on composite degradations in real datasets, **we have addressed the issue through additional experiments and analyses, and the reviewer has subsequently agreed to raise the score to 5**.
> > > > >
> > > > > For Reviewer iFAK’s unfamiliarity with certain metrics, we have further detailed all the evaluation metrics used in our work to demonstrate the overall soundness of our assessment framework. Notably, these metrics are among the most widely used in the image fusion field [1,2].
> > > > >
> > > > >
> > > > > Finally, please carefully review our previous responses; therefore, we do not repeat detailed answers here.
> > > > >
> > > > > Once again, we sincerely appreciate your feedback and hope our replies have addressed all your concerns. If you have further questions or suggestions, **please specify them** and feel free to contact us. We remain patient and look forward to your feedback at any time.
> > > > >
> > > > >
> > > > > [1] Cao, Bing, et al. Conditional controllable image fusion. NeurIPS 2024.
> > > > >
> > > > > [2] Zhao, Zixiang, et al. Equivariant multi-modality image fusion. CVPR 2024.

---

> ### Author Response · Authors · 2025-08-08
>
> First, we appreciate that our response has partially addressed some of your concerns. For the remaining points, please allow us to offer further clarification; we hope this will be helpful.
>
> **1. Clarification of the MICP mechanism**
>
> First, it should be clarified that the Laplacian operator is used only in the second projection stage of the MICP mechanism. The formula is (Appendix Eq. (3)):
>
> $\min_{\mathcal{D}^d} \left\| \nabla \left( \mathcal{D}^s + \mathcal{D}^d \right) - \max\left( \nabla i^m, \nabla i^n \right) \right\|_2^2 + \phi \left\|\nabla^2 \mathcal{D}^d \right\|_2^2 $
>
> Here, we **optimize to make the Laplacian gradient $\nabla^2$ smaller to suppress grid artifacts caused by over-optimization of the detail layer $\mathcal{D}^d$, rather than to enhance texture.** The primary texture-preserving constraint is the first term, which enforces preservation of the maximal gradients across source images; this is consistent with the maximal texture-gradient loss commonly used in image fusion.
>
> Moreover, our comparative and ablation experiments on the same dataset show that PDFuse-B, which does not integrate the interference operator set, lacks degradation-removal capability. PDFuse-B focuses on improving fusion quality rather than degradation removal and does not show substantial gains in EI, SF, or AG. This indicates that the degradation removal observed in PDFuse-E, and the marked improvements in EI, AG, and SF, are not attributable to the Laplacian term in the MICP mechanism.
>
> Finally, the appendix and discussion provide detailed derivations of the MICP mechanism， from the loss formulation to the two-stage projection process， and a proof of convergence for the projection optimization. If any specific derivation steps remain unclear, **please indicate which specific steps are unclear**, and we will be glad to provide further explanations.
>
> **2. Clarification on runtime-efficiency experiments**
>
> We report runtime statistics for the compared methods in Appendix D (Limitations). We acknowledge that, compared with non-diffusion methods, our approach has an efficiency limitation. However, as the number of input images increases, our runtime remains nearly constant, while that of others grows roughly linearly.
>
> Compared with DDPM-based methods (CCF and DDFM), we analyzed different diffusion-step settings and followed each method’s default step count. With fewer steps, our method already achieves better performance. Moreover, even **at the same number of diffusion steps**, our method requires at most **0.64×** the runtime of CCF/DDFM.
>
> We attribute this advantage to architectural differences. LDMs operate on a latent space whose U-Net input spatial scale is 1/64 that of a DDPM U-Net, and the LDM VAE contains fewer convolutional and residual blocks than a DDPM U-Net. By contrast, DDPMs perform many ResBlock+Attention operations directly in pixel space on large images, which is much more time-consuming. Therefore, despite the VAE decode step required in each regularization step of LDMs, our method remains more time-efficient.
>
> **3. Clarification on the degradation-operator set**
>
> The operator set we use covers common degradations in multimodal image fusion, including low light, overexposure, underexposure, blur, and thermal diffusion, which we endeavored to include as comprehensively as possible. We also evaluated several composite degradations (for example, low light combined with blur) and found that our method can effectively handle such combinations. Notably, most supervised degradation datasets for multimodal image fusion construct complex composite degradations by stacking multiple simple degradation types.
>
> Furthermore, in Appendix B.2, we show results on various real datasets with unknown degradations. Despite the unknown nature of these degradations, integrating the interference-operator set improves the fused images by suppressing certain degraded features and enhancing overall quality (see Appendix Fig. s1–s4).

---

### Official Review · Reviewer_iFAK · 2025-06-20

**Clarity:** 1
**Significance:** 2
**Originality:** 2
**Rating:** 4
**Confidence:** 2

**Summary:**

This paper introduces PDFuse, a training-free image fusion framework built on pre-trained latent diffusion models with a novel projection–manifold regularization strategy.

By formulating fusion as a constrained diffusion inference process, it effectively integrates multiple source images to generate high-fidelity results.

Two core mechanisms, Multi-source Information Consistency Projection (MICP) and Latent Manifold-preservation Guidance (LMG), ensure both consistency with source inputs and alignment with the generative prior.

Extensive experiments show that PDFuse delivers competitive performance across diverse image fusion tasks without requiring supervised training.

**Questions:**

### Questions for the Authors

**Q1:** Lines 61–62 state that "PDFuse constructs a canonical interference operator set."
- What exactly is meant by a *canonical interference operator set* in this context?
- Why is this set necessary, as mentioned again in Line 173?
- Please clarify its role in the method and consider adding an ablation or analysis to demonstrate its impact.

**Q2:** The paper heavily emphasizes fusion from multiple source images (e.g., `i_m`, `i_n`), but **Figure 1** and **Figure 2** do not visually explain how these sources are processed or interact during inference.
- Can the authors revise or augment the diagrams to clearly depict the role and flow of multiple inputs?

**Q3:** The qualitative results in **Figures 3, 5, and 6** do not convincingly show improvements of the proposed method.
- Could the authors enhance the visualization quality and include zoomed-in or region-focused comparisons to better highlight differences?
- Also, **Figure 5** is not referenced or discussed in the text—please explain its purpose and relevance.

**Q4:** The paper includes many tables and figures, but lacks clear conceptual takeaways or technical insights.
- Can the authors restructure the results section to highlight key observations and explain what we learn from each experiment, rather than simply reporting numbers?

**Q5:** The ablation study is quite shallow and lacks detailed analysis.
- Can the authors expand this section to better isolate the contributions of individual components (e.g., MICP, LMG, interference operators) with clear comparisons?

**Q6:** In **Table 2**, the comparison with **EAT (TMM’25)** is inconclusive due to missing metrics.
- Can the authors either reproduce the missing metrics for a fair comparison or discuss this limitation more clearly to help contextualize their method’s relative performance?

**Ethical Concerns:**

["NO or VERY MINOR ethics concerns only"]

**Final Justification:**

After reviewing it, I find that it partially addresses my concerns. The proposed method presents some meaningful contributions, particularly the integration of the interference operator set and the detailed ablation studies. However, the overall clarity and organization of the paper remain challenging to follow. Given these considerations, I slightly raise my score to reflect the improvements. Additionally, some of the metrics introduced by the authors are unfamiliar to me, so I have lowered my confidence and will defer to other reviewers and the ACs for the final judgment.

**Limitations:**

Yes

**Quality:**

2

**Strengths And Weaknesses:**

**Strengths**
- The submission is complete, and the writing is generally clear and accessible to readers.
- The method is training-free, which reduces computational cost and simplifies deployment.
- Some quantitative results demonstrate the potential of the proposed approach.

**Weaknesses**
- The paper emphasizes the role of multiple source images (e.g., `i_m`, `i_n`), yet the visual diagrams (e.g., **Figure 1** or **Figure 2**) do not illustrate how these sources interact with the fusion process, making the method less intuitive to understand.
- The qualitative results are not convincing, **Figures 3, 5, and 6** do not clearly show visual improvements from the proposed method, and **Figure 5** appears unexplained or unreferenced in the main text.
- The paper feels overloaded with tables and figures, but lacks clear insights or takeaways. The **ablation study** section is shallow and does not offera  meaningful understanding of the proposed components or their contributions.
- In **Table 2**, comparisons to baselines like **EAT (TMM’25)** are incomplete due to missing metrics, making it difficult to assess the true advantage of the proposed method.

---

> ### Author Rebuttal · Authors · 2025-07-31
>
> **Q1**: Analysis of the canonical interference operator set.
>
> **Reply**: Multi-modal image fusion is often limited by scene degradations, hindering the effective integration of advantages from different modalities. To improve the robustness of fusion methods against such degradations, we constructed **a canonical interference operator set**, consisting of a series of explicit models for common degradations such as low light, improper exposure, blur, and thermal diffusion.
>
> Inspired by PSLD (NeurIPS 2023), we incorporated this operator set into the regularization processes of the MICP and LMG mechanisms, utilizing diffusion priors to address these degradations. Specifically, the two-stage projection of MICP, as defined in Eq. (8) and Eq. (10), is modified by the interference operator set, as shown in Eq. (14) and Eq. (15). Similarly, the LMG mechanism, as in Eq. (12), is adjusted after integrating the operator set, as shown in Eq. (16). This integration allows the diffusion process to address the inverse problem of the degradation operator $\mathbf{\Pi}$, effectively mitigating various degradations in multi-modal image fusion. To validate the effectiveness of the canonical interference operator set, we evaluated its de-degradation capability in challenging scenarios, as illustrated in Fig. 4 and Table. 1. In the ablation experiments, we compared PDFuse-B (without the interference operator set) and PDFuse-E (with the interference operator set) in Fig. 8 and Table 5. The experimental results demonstrate that PDFuse-E, which integrates the interference operator set, more effectively handles degradations such as low light and blur, outperforming PDFuse-B in the multi-modal fusion process.
>
> **Q2**: Enhance the intuitiveness of the information interaction process across multiple sources.
>
> **Reply**: We sincerely appreciate the improvement suggestions you have provided. Specifically, we have designed two efficient multi-source image regularization mechanisms to constrain the diffusion process. These mechanisms first extract key information, such as coarse structure, pixel intensity, and texture, from the multi-source images through information metrics. Then, they inject this information into the diffusion process through projection and gradient constraints.
> The MICP mechanism ensures the information fidelity of the fused image through two-stage projection regularization. On the other hand, the LMG mechanism simultaneously constructs gradients between $\hat{\mathbf{z}}_0^t$ and the multi-modal images in both pixel and latent spaces, guiding the diffusion process to generate high-quality fusion results.
> In the camera ready, we commit to incorporating intuitive representations of these interactions and expand the framework diagram as in Fig. 2 to more clearly illustrate the multi-modal information interaction process.
>
> **Q3**: The visualization of the qualitative results in Fig. 3, Fig. 5, and Fig. 6, ​​with an explanatory note on Fig. 5​.
>
> **Reply**: Fig. 5 presents the qualitative results for multi-exposure image fusion. We discussed these results in line 218 of the main text, but we are sorry for mistakenly referring to Fig. 4. We provide several explanations to help convey the visual advantages of our method.
>
> (1) Fig. 3 presents the qualitative results for infrared-visible image fusion. According to the qualitative analysis, our method exhibits superior infrared salience in pedestrians and effectively preserves fine details in both infrared and visible textures (e.g., bicycle and tree trunk textures).
>
> (2) Fig. 5 presents the qualitative results for multi-exposure image fusion. Our method shows a significant advantage in overall exposure levels and color contrast compared to other methods. Additionally, the highlighted regions confirm the superiority of our method in preserving textures in different exposure images (e.g., windows, posters, clouds).
>
> (3) Fig. 6 presents the qualitative results for multi-focus image fusion. Our method is capable of preserving textures across different focus depths while maintaining a high overall image contrast. Specifically, compared to the similar diffusion method CCF, our method shows a significant improvement in texture fidelity.
> Thanks for pointing out this issue. We will carefully correct all typos in the final version and further include zoomed-in or region-focused comparisons to enhance the presentation of the figures and tables.
>
> **Q4**: Reorganize the results section and provide specific analysis.
>
> **Reply**: **Reply**: We commit to reorganizing the results section to present a deeper theoretical analysis in the camera ready. Here, we attempt to analyze the experiments in conjunction with the underlying principles of our method to provide valuable insights.
>
> (1) We evaluated our method on VIF, MEF, and MFF tasks. Compared to the diffusion-based CCF of the same class, our approach demonstrates superior texture preservation, particularly in MFF in Fig. 6 and Table. 3. This stems from key differences in regularization. CCF guides the DDPM process using pixel-level gradients, where large gradients risk over-shifted results, necessitating small, multi-step updates for fidelity. In contrast, our method employs the MICP mechanism with a multi-stage dual projection strategy, enabling hierarchical, convergent optimization of coarse structures $\mathcal{D}^s(\hat{\mathbf{z}}_0^t)$ and fine details $\mathcal{D}^d(\hat{\mathbf{z}}_0^t)$ while mitigating conflicts. Consequently, our method achieves more effective texture retention than CCF.
>
> (2) Furthermore, as shown in the qualitative results (Figs. 3, 5, and 6), our method achieves superior contrast and color fidelity across tasks compared to other fusion approaches, and in some cases even surpasses the multi-source images. This is further supported by the higher SD values reported in Tables. 1, 2, and 3. These enhancements result from the combined effect of the MICP and LGM mechanisms. MICP enables efficient and high-fidelity preservation of multi-source information, and LGM guides the latent variable $\hat{\mathbf{z}}_0^t$ to maintain diffusion consistency with the source images through gradient-based regulation. These allow the generative model to enhance overall image quality in terms of contrast and color, while simultaneously injecting source image information, thereby achieving superior performance in these aspects.
>
> **Q5**: A more detailed analysis of the ablation experiments.
>
> **Reply**: In the ablation study, we evaluated the removal of the LMG mechanism (Method I), the first stage of MICP (Method II), the second stage of MICP (Method III), the complete MICP mechanism (Method IV), as well as variants without (PDFuse-B) and with (PDFuse-E) the interference operator set.
>
> The results reveal that removing the first projection stage of the MICP mechanism leads to color distortion, as this projection  stage is responsible for injecting the coarse structural component $\mathcal{D}^s(\hat{\mathbf{z}}_0^t)$ into the diffusion process. Without it, low-frequency and color information cannot be effectively preserved. Removing the second projection stage results in significant texture loss, since it handles the injection of fine detail components $\mathcal{D}^d(\hat{\mathbf{z}}_0^t)$ into the diffusion process. When the entire MICP mechanism is removed, relying solely on the LMG mechanism fails to maintain information fidelity, leading to poor fusion quality. Conversely, when the LMG mechanism is removed, the MICP mechanism still supports structural and texture preservation. However, compared to PDFuse-B (which includes the LMG mechanism), the fusion results show reduced contrast and darker tones due to the absence of manifold-preservation guidance from the LMG mechanism, resulting in limited use of generative priors. Finally, PDFuse-E, which integrates the interference operator set, outperforms PDFuse-B by leveraging diffusion priors to solve inverse degradation problems. It effectively handles low light, blur, and other distortions, yielding better overall visual quality.
>
> We commit to including these detailed ablation analyses of the MICP mechanism, LMG mechanism, and the interference operator set in the final version to better emphasize the contributions of each component.
>
> **Q6**: Additional metrics are introduced and compared with EAT.
>
> **Reply**: Due to the lack of ground truth (GT) in image fusion tasks, existing methods largely adopt unsupervised metrics and some reference-based metrics for performance evaluation. Our method employs evaluation metrics including SF, EI, AG, SD, SCD, and Q_cb, which are widely used in mainstream approaches (such as the selected competitors: CCF, TC‑MoA, and HSDS‑MEF).  Notably, the metrics used in EAT are not the standard ones commonly adopted in multi‑exposure image fusion (MEF). To enhance the comparison comprehensiveness, we additionally introduce **MEF‑SSIM** (IEEE TIP 2015) for evaluation. MEF‑SSIM is a structural similarity evaluation metric specifically designed for MEF, which has been widely adopted and recognized in the MEF field. This metric performs perception‑driven quality assessment by measuring the consistency between the fusion result and the ideal structures of multi‑source exposure images. We follow the setup of the main text and conduct tests on the MEFB dataset, with the implementation code obtained from MEFNet (IEEE TIP 2019). The quantitative experimental results are shown below:
>
> | Method|U2Fusion|DeFusion|HSDS‑MEF|TC‑MoA |CCF|SAMT|EAT|**PDFuse (Ours)**|
> |-|-|-|-|-|-|-|-|-|
> |**MEF‑SSIM**|0.8030|0.8297|0.8164|0.8838|0.7963|0.8617|0.8574|**0.8856**|
>
> As shown, our method still achieves the highest performance under the MEF‑SSIM metric, further validating its fusion quality effectiveness and advancement.

---

> > ### Comment · Reviewer_iFAK · 2025-08-05
> >
> > Thanks for sharing the rebuttal. After reviewing it, I find that it partially addresses my concerns. The proposed method presents some meaningful contributions, particularly the integration of the interference operator set and the detailed ablation studies. However, the overall clarity and organization of the paper remain challenging to follow. Given these considerations, I will slightly raise my score to reflect the improvements. Additionally, some of the metrics introduced by the authors are unfamiliar to me, so I have lowered my confidence and will defer to other reviewers and the ACs for the final judgment.

---

> > > ### Author Response · Authors · 2025-08-07
> > >
> > > We sincerely appreciate the valuable time and thoughtful consideration you have dedicated to reviewing this work.
> > >
> > > We are pleased that the detailed clarifications we provided have addressed some of your concerns, and we are also delighted to receive your recognition of the significance of our contributions (particularly the interference operator set and the detailed ablation study). For the two remaining concerns, we would like to offer one final explanation and clarification, which we hope will be valuable to you:
> > >
> > > 1. First, regarding the clarity and structure of the manuscript, we are grateful to the reviewer for the series of improvement suggestions, which have been invaluable in enhancing the overall quality of the paper. We again commit that, in the final version, we will incorporate the above feedback as follows:
> > >    a. We will present the multi-source information interaction process between the MICP and LMG mechanisms in the framework diagram in a more intuitive manner.
> > >
> > >    b. We will enlarge the visual results for qualitative comparisons wherever possible to improve clarity.
> > >
> > >    c. We will organize all details of the comparative experiment analyses and ablation study analyses from the discussion section and integrate them into the main text.
> > >
> > >    d. We will merge the main text and the appendix, reformatting sections of both into a unified layout and using the appendix to further elaborate on each regularization mechanism and the interference operator set, thereby doing our utmost to enhance overall clarity.
> > >
> > > Hence, we hope this will resolve your remaining concerns regarding the manuscript’s clarity.
> > >
> > > 2. Secondly, regarding the evaluation metrics, we would like to briefly restate the seven indicators we adopted and the rationale for their selection. Image fusion seeks to combine informative content from multiple source images but typically lacks ground truth for the fused output. To address this, we use seven metrics: EI, SF, EN, AG, SD, SCD, TE, and Qcb, forming a unified evaluation framework.
> > >
> > >
> > >    **Non-reference metrics** assess various intrinsic properties of the fused image:
> > >    - **EI**  measures the overall strength of edge textures in the fused image, reflecting its structural contour integrity.
> > >    - **SF** evaluates the fused image’s spatial activity by aggregating the strength of its frequency components.
> > >    - **AG**  focuses on local intensity changes to quantify the level of detail in the fused image.
> > >    - **EN**  uses information entropy to gauge the amount of information contained in the fused image.
> > >    - **SD**  measures the overall contrast of the fused image using standard deviation.
> > >
> > >
> > >    Together, these five metrics provide a comprehensive assessment—from structural contours and frequency content to local detail, information richness, and contrast—of the fused image’s quality.
> > >
> > >    **Reference metrics** (require access to the source images) evaluate how well the fused image preserves and relates to the inputs:
> > >    - **SCD** computes the correlation between the difference image of each source and the fused result to assess fusion quality.
> > >    - **TE**  reflects the degree of information retention by measuring the information discrepancy between each source and the fused image.
> > >    - **Qcb** captures similarity in local contrast between the fused image and the source images, indicating how well structural details are preserved.
> > >
> > >    By combining these, we jointly evaluate correlation, information preservation, and structural similarity between the fusion result and its multiple inputs.
> > >
> > >    These seven metrics are among the most widely adopted in the image fusion community [1,2].
> > >
> > >    In our discussion section, we also introduce **MEF-SSIM**, a reference metric designed for multi-exposure fusion tasks. MEF-SSIM assesses fused vs. source images across multiple scales, capturing consistency in luminance, contrast, and structural information specifically for multi-exposure fusion.
> > >
> > > We hope this concise overview clarifies the choice and appropriateness of our evaluation metrics.
> > >
> > > Once again, we sincerely appreciate your constructive feedback and the guidance you’ve provided to improve our work. Should you have any further questions or wish to discuss any aspect in more detail, please do not hesitate to contact us; I remain available at all times.
> > >
> > > [1] Cao, Bing, et al. "Conditional controllable image fusion." Advances in Neural Information Processing Systems 37 (2024): 120311-120335.
> > >
> > > [2] Zhao, Zixiang, et al. "Equivariant multi-modality image fusion." Proceedings of the IEEE/CVF conference on computer vision and pattern recognition. 2024.

---

> ### Author Response · Authors · 2025-08-05
>
> We sincerely appreciate your insightful and valuable comments, which have significantly contributed to improving the overall quality of our paper. In response to your concerns and suggestions, we have provided detailed clarifications and explanations regarding the canonical interference operator set, the information interaction process across multiple sources, the visualization of qualitative results, the results section, and the ablation experiments. In addition, we have included comparative results on the MEF-SSIM metric between our method and other advanced approaches such as EAT. We promise that all the above revisions will be incorporated into the camera-ready version without reservation.
>
> We hope we have addressed all of your concerns. Please feel free to let us know if you have any additional questions or suggestions. We will address them promptly and make every effort to resolve them.

---

### Official Review · Reviewer_8Yc3 · 2025-06-25

**Clarity:** 3
**Significance:** 3
**Originality:** 3
**Rating:** 5
**Confidence:** 4

**Summary:**

This paper proposes **PDFuse**, a training-free, general-purpose, and degradation-robust image fusion framework based on pretrained latent diffusion models. The key idea is to reformulate image fusion as a posterior sampling process guided by two regularization mechanisms:
- **MICP (Multi-source Information Consistency Projection)**: injects multi-source image content via a two-stage projection strategy.
- **LMG (Latent Manifold-preservation Guidance)**: enforces generative consistency by aligning latent variables with the manifold prior.

To further enhance robustness, a set of degradation operators is embedded into both mechanisms, allowing PDFuse to adapt to various degradation scenarios. Extensive experiments across diverse fusion tasks (IVF, MEF, MFF) and semantic segmentation verify the superiority and robustness of the proposed approach.

**Questions:**

- How does the method scale to fusion tasks involving more than two source images?
- Can the proposed degradation operators generalize to unseen types or compositions?

**Ethical Concerns:**

["NO or VERY MINOR ethics concerns only"]

**Final Justification:**

The response has addressed my concerns, for which I lean towards acceptance of this paper.

**Limitations:**

See above.

**Paper Formatting Concerns:**

none.

**Quality:**

3

**Strengths And Weaknesses:**

### **Strengths**

1. **Well-engineered and effective framework**
   The alternating design of MICP and LMG offers a principled mechanism to balance fidelity and generative quality, and integrates seamlessly into latent diffusion inference.

2. **Training-free and broadly applicable**
   The method does not require retraining and generalizes across tasks and degradations, making it practically useful. The integration of an interference operator set further improves robustness.

3. **Comprehensive experiments**
   Strong empirical results on multiple tasks, supported by ablation studies and semantic-level verification, demonstrate the method's effectiveness and generalization.

---

### **Weaknesses**

1. **Limited theoretical novelty**
   The approach is largely driven by system design and heuristics. There is no formal analysis of the convergence, optimality, or theoretical behavior of the alternating regularization scheme.

2. **Somewhat heuristic projection and loss design**
   The two-stage MICP and the handcrafted LMG loss rely on pre-defined forms (e.g., max intensity, Laplacian regularization), which may limit adaptability beyond the chosen tasks.

3. **Lack of analysis on degradation generalization**
   The degradation operator set is conceptually flexible, but it’s unclear how well it handles unseen or compound degradations at test time.

---

> ### Author Rebuttal · Authors · 2025-07-30
>
> **Q1**: Analysis of the convergence, optimality of the alternating regularization scheme.
>
> **Reply**: Our method defines image fusion as a latent diffusion process with multi-source regularization constraints, which is the first use of pre-trained LDMs for a training-free, general, and robust fusion approach.
>
> **Convergence and Optimality Analysis:**
>
> (1) The MICP mechanism regularizes the diffusion process by projecting the latent intermediate variable $\hat{\mathbf{z}}_0^t$ in two stages: first, the projection of the coarse structural component to obtain $\mathcal{D}^s(\hat{\mathbf{z}}_0^t)$, followed by the projection of the fine detail component $\mathcal{D}^d(\hat{\mathbf{z}}_0^t)$. This two-stage approach mitigates gradient conflicts that arise from optimizing multiple objectives simultaneously. Notably, both projection stages can be solved using conjugate gradient iterative modeling. In Appendix A.1, we analyze the coefficient matrices for the conjugate gradient solution of the two-stage projection. It has been proven to be symmetric and positive definite, ensuring strict convexity. Therefore, the MICP mechanism is convergent, and the optimal solution is guaranteed within a finite number of iterations.
>
> (2) The LMG mechanism enforces consistency between $\hat{\mathbf{z}}_0^t$ and multiple source images through pixel-level and latent space-level regularization. The pixel-level regularization term is derived from widely used loss functions in the image fusion field **(MURF，TPAMI 2023；SHIP，TPAMI 2024)**. However, the many-to-one mapping in VAE decoding can lead to multiple optimization directions when relying solely on pixel-level gradients. Latent space-level regularization resolves this issue, guiding $\hat{\mathbf{z}}_0^t$ toward a more definitive optimization direction, thus improving convergence. Despite this, because the loss function involves multi-objective optimization and is non-convex, it often leads to convergence at local optima.
>
> (3) Given that image fusion tasks lack ground truth (GT), multiple quantitative metrics and human visual assessment are used for performance evaluation. Additionally, given the complexity of posterior sampling in the diffusion process, it is challenging to apply specific theoretical tools to guarantee optimal fusion performance of alternating optimization. However, we have validated the effectiveness of the alternating optimization regularization mechanisms as thoroughly as possible through ablation experiments. In the ablation study presented in the main text, we evaluated the first projection stage of the MICP mechanism (Method II), the second stage (Method III), and the LMG mechanism (Method I) from both qualitative and quantitative perspectives. The results clearly show that the method with alternating optimization outperforms any individual regularization mechanism, fully validating its effectiveness.
>
>
>
> **Q2**: The adaptability of the projection and loss design.
>
> **Reply**: The two fundamental elements of an image are pixel intensity and gradient, representing contrast distribution and texture structure, respectively. In the multi-source image fusion, we aim to maintain significant contrast while preserving the richest texture details. This concept is applicable to most image fusion tasks. In our projection setup, the overall contrast distribution is projected based on the pixel intensity of the fusion scene, adapting to various fusion scenarios. For instance, in infrared-visible image fusion, we aim to capture the maximum intensity to ensure the saliency of thermal objects while preserving the texture of visible modality to the greatest extent. For multi-exposure image fusion, our focus is to ensure the fused image's overall contrast aligns with the desirable exposure level while maximizing the integration of scene structures. Our adopted loss based on pixel intensity and gradient is widely recognized and used in the image fusion field (MURF, TPAMI 2023; SHIP, TPAMI 2024).
>
> To evaluate the broader applicability of the proposed method across various image fusion tasks, we also test its performance in multi-polarization image fusion, which integrates imaging data from different polarization angles to fully capture the scene's polarization. The quantitative results are as follows:
>
> |Method|EI↑|SF↑|EN↑|AG↑|SD↑|TE↓|SCD↑|Q_cb↑|
> |-|-|-|-|-|-|-|-|-|
> |U2Fusion|24.573|8.279|5.143|2.342|14.887|6.886|1.280|0.418|
> |DeFusion|28.454|8.178|6.448|2.616|31.290|5.580|1.543|0.353|
> |DDFM|37.094|12.696|6.518|3.649|30.066|5.513|1.884|0.384|
> |EMMA|39.968|11.703|6.753 |3.704|38.674|5.276|1.699|0.387|
> |TC-MoA|43.186|14.358|6.646|4.187|34.115|5.382|1.768|0.387|
> |CCF|45.828|17.102|6.382|4.931|25.846|5.646|1.671|0.382|
> |DCEvo|43.249|14.682|6.703|4.193|36.551|5.326|1.771|0.413|
> |PDFuse|**48.941**|**15.687**|**6.796**|**4.681**|37.523(#2)|**5.232**|1.808(#2)|0.389(#3)|
>
> In multi-polarization image fusion, our method still excels in texture details, contrast, correlation, and structural consistency, showcasing strong generalization performance.
>
> **Q3**: Extending to fusion tasks involving more than two source images
>
> **Reply**: Since our method does not impose an explicit limit on the number of inputs. Consequently, our approach can process an arbitrary number of source images in a single pass with negligible additional computational or temporal overhead. When handling more than two source images, the MICP mechanism concurrently analyzes the pixel- and texture-level details of all inputs. It injects them into the diffusion process via its two-stage projection. Similarly, the LMG mechanism establishes gradient‑based constraints across all source images. The underlying principles and workflow of the mechanisms remain unchanged; it only requires increasing the number of information sources within the mechanisms.
> We validated this capability using the SICE multi-exposure dataset by inputting three images with varying exposure levels. It is worth emphasizing that other methods can only process two images at a time, so they perform multi-exposure fusion of three images in two separate stages via sequential pairing. In contrast, our method achieves this in a single pass. The quantitative results are presented below:
>
> |Method|EI↑|SF↑|EN↑|AG↑|SD↑|TE↓|SCD↑|Q_cb↑|
> |-|-|-|-|-|-|-|-|-|
> |U2Fusion|85.165|22.561|7.225|8.048|54.56|13.25|0.716|**0.500**|
> |DeFusion|67.214|17.257|7.279|6.206|52.841|13.201|0.550|0.357|
> |HSDS-MEF|95.637|29.464|6.994|9.367|73.821|13.486|2.989|0.351|
> |TC-MoA|89.575|24.851|7.532|8.611|57.462|12.947|0.959|0.466|
> |CCF|86.120|28.723|7.614|8.885|**70.408**|12.866|**3.575**|0.461|
> |SAMT-MEF|89.485|29.514|7.425|9.378|58.071|13.055|1.225|0.430|
> |EAT|36.849|9.384|6.501|3.146|40.089|13.937|1.700|0.347|
> |**PDFuse**  |**112.422**|**30.975**|**7.708**|**10.930**|68.581(#2)|**12.771**|2.349(#3)|0.489(#2)|
>
> From the experimental results, although our method requires only a single regularized diffusion step to process three multi-exposure images—whereas other methods must perform multiple sequential fusion stages—our approach still achieves leading performance on most evaluation metrics (#1 or #2), demonstrating that our regularization mechanism can be generally applied to fuse three or more source images. In addition, we further analyzed how the runtime and the Edge Intensity (EI) of the fused images produced by our method, PDFuse, change as the number of input images with different exposure settings varies. The quantitative results are presented in the table below:
>
> | Input sequence count |2|3|4|5|6|7|8|9|
> |-|-|-|-|-|-|-|-|-|
> | EI↑|95.156|112.42|117.81|119.52|121.84|121.86|123.01|**124.45**|
> | Runtime (s)|18.08|18.15|18.03|18.24|18.33|18.19|18.30|18.23|
>
> Quantitative analysis shows that as the number of input images with varying exposure settings increases, the scene texture (EI) of the fused images improves, while the runtime remains largely unaffected.
>
> **Q4**: Handle compound and invisible degradations
>
> **Reply**: The flexible interference operator set we designed accommodates various degradations in multimodal image fusion. In the actual design,  we have aimed to incorporate as many degradation types as possible, such as low light, exposure, blur, and thermal diffusion. Notably, our method is not limited to handling single degradation types but can also address composite degradations. For example, in the main text, we demonstrate the challenging scenario of low light and blur composite degradation in Fig. 3, where our method effectively mitigates the degradation, showcasing its superior fusion capabilities. To further validate its generalization, we evaluated our method on several real degradation datasets, in Appendix B (Fig.s1, Fig.s2, Fig.s3, Fig.4), which include both well-defined degradations, such as low light, improper exposure, and blur, as well as more complex, unseen degradations like low contrast, fog, glare, and compression losses that were not explicitly defined in the interference operator set. We analyze that the handling of composite degradations can be achieved by combining and integrating single degradations into the regularization mechanism. Although real, unseen degradations may not be precisely modeled, they often contain already defined degradation types or share similarities with defined degradations, such as low contrast, potentially containing some improper exposure.
>
> In general, the flexible compatibility of our method with the interference operator set enables it to effectively handle composite degradations, including previously unseen types. Furthermore, we would like to emphasize that our method is the first to utilize a pre-trained latent diffusion model (LDM) for unsupervised degradation handling in image fusion. By leveraging the LDM, it can recover texture details lost due to degradation, achieving satisfactory performance. This approach offers new insights for research on unsupervised robust fusion methods.

---

> > ### Comment · Reviewer_8Yc3 · 2025-08-05
> >
> > Thank you for the detailed response, which has addressed most of my concerns. However, since the authors claim that PDFuse can effectively handle composite degradations, including previously unseen types, it would be more convincing to evaluate the method under real‑world scenarios or clarify its advantages over state‑of‑the‑art methods [1,2] tailored to this problem. For example, testing on composite weather degradations such as haze and rain, as considered in [1,2], would strengthen the claim.
> >
> > Overall, I lean towards acceptance of this paper. The rating can be increased to 5 if the claim is further justified in the revision.
> >
> > [1] Tang, X., Gu, X., He, X., & Sun, J. Degradation‑Aware Residual‑Conditioned Optimal Transport for Unified Image Restoration. IEEE TPAMI, 2025.
> >
> > [2] Guo, Y., Gao, Y., Lu, Y., Liu, R. W., & He, S. OneRestore: A Universal Restoration Framework for Composite Degradation. ECCV 2024.

---

> > > ### Author Response · Authors · 2025-08-07
> > >
> > > We sincerely appreciate the valuable time and thoughtful consideration you have dedicated to reviewing this work.
> > >
> > > We are pleased that our detailed clarifications have resolved most of your concerns, and we appreciate your explicit recommendation to accept this paper. In response to your suggestion to evaluate PDFuse-E under complex weather degradation scenarios, we have carefully reviewed the two image-restoration references you provided and present the following analysis and experiments:
> > >
> > > Reference [1] employs a degradation-aware residual conditional optimal transport framework for unified degradation modeling. Reference [2] uses a Transformer architecture with text prompts and visual-feature discrimination to handle multiple degradation types. Both rely on supervised datasets containing rain, fog, low-light and other degradations to train a single model capable of addressing individual and composite degradations at test time. These methods are designed for restoring single-modal visible-light images. In contrast, Our approach targets multimodal image fusion. By integrating the interference operator set into the fusion regularization mechanism without relying on degradation-specific training data, we implicitly unify fusion and degradation removal, suppressing degradation in the multimodal inputs and preventing its influence on the information metrics of the fusion process. Therefore, the supervised degradation datasets used in [1] and [2] are not suitable for evaluating our method.
> > >
> > >
> > > To this end, we extracted challenging complex-weather degradation scenes from the M3FD infrared–visible image fusion dataset [3], captured with a dual‐lens optical camera and a dual‐lens infrared sensor. The dataset includes adverse conditions such as haze + low‐light, low‐light + headlight glare, and haze + noise. We compared our PDFuse-E method against two cascaded schemes:
> > > 1. **Scheme 1:** Pre‐enhance the visible image using the DA-RCOT restoration method from [1], then fuse with DCEvo.
> > > 2. **Scheme 2:** Pre‐enhance the visible image using the OneRestore method from [2], then fuse with DCEvo.
> > > The quantitative comparison results are shown below:
> > >
> > > | Method                  | EI     | SF     | AG     | EN    | SD     | TE    | SCD   | Qcb   |
> > > |-------------------------|--------|--------|--------|-------|--------|-------|-------|-------|
> > > | **Metric Type**      | Non-reference | Non-reference | Non-reference | Non-reference | Non-reference | Full-reference | Full-reference | Full-reference |
> > > | DA‑RCOT + DCEVo        | 67.56  | 17.94  | 6.31   | 6.78  | 32.71  | 6.89  | 1.57  | 0.51  |
> > > | OneRestore + DCEvo      | 105.45 | 27.90  | 9.86   | 7.14  | 42.20  | 6.53  | 0.96  | 0.45  |
> > > | PDFuse‑E                | 96.07  | 24.54  | 9.08   | 6.88  | 35.14  | 6.79  | 1.35  | 0.49  |
> > >
> > >
> > > From the table, we observe that the OneRestore + DCEvo method achieves the best performance on several non-reference metrics such as EI, SF, and AG, but yields relatively low scores on full-reference metrics like SCD and Qcb. In contrast, the DA-RCOT + DCEvo method performs best on SCD and Qcb, yet falls behind on non-reference metrics such as EI and SF.
> > >
> > > While our method does not outperform the others on any single metric, it achieves a more balanced overall performance. For example, our scores on non-reference metrics like EI and AG are very close to those of OneRestore + DCEvo and significantly higher than those of DA-RCOT + DCEvo. At the same time, our full-reference scores on SCD and Qcb remain relatively high.
> > >
> > > From a qualitative perspective, in low-light and extreme haze-degraded scenes, both OneRestore and DA-RCOT struggle to restore visibility in the hazy regions of the visible image, resulting in fused outputs that retain heavy haze and hinder the integration of infrared information. While our PDFuse-E method also cannot directly restore haze in the visible image, the interference operator set adjusts the global exposure structure, effectively suppressing haze brightness and thus reducing its fusion weight. This enables the fused output to preserve rich infrared texture information in those regions.
> > >
> > > In the final version, we will include the above quantitative results in the appendix and provide qualitative visual comparisons to more intuitively highlight and discuss the limitations and advantages of our method under complex degradation scenarios.
> > >
> > > Once again, we appreciate your valuable feedback and guidance. Please feel free to contact us with any further questions. We are always available.
> > >
> > > [1] Tang, X., Gu, X., He, X., & Sun, J. Degradation‑Aware Residual‑Conditioned Optimal Transport for Unified Image Restoration. IEEE TPAMI, 2025.
> > >
> > > [2] Guo, Y., Gao, Y., Lu, Y., Liu, R. W., & He, S. OneRestore: A Universal Restoration Framework for Composite Degradation. ECCV 2024.
> > >
> > > [3] Liu, Jinyuan, et al. Target-aware Dual Adversarial Learning and a Multi-scenario Multi-Modality Benchmark to Fuse Infrared and Visible for Object Detection. CVPR 2022.

---

> ### Comment · Reviewer_8Yc3 · 2025-08-08
>
> Thank you for the additional results, which provide strong evidence for the practical effectiveness of PDFuse in addressing real-world composite degradations. Accordingly, I am increasing my rating to 5.

---

### Official Review · Reviewer_CeyL · 2025-06-29

**Clarity:** 3
**Significance:** 3
**Originality:** 4
**Rating:** 5
**Confidence:** 5

**Summary:**

This paper proposes a training-free image fusion framework built on a pre-trained latent diffusion model. This framework presents two regularization mechanisms: Multi-Source Information Consistency Projection (MICP) and Latent Manifold-preservation Guidance (LMG). Then, the unified image fusion modeling is realized through the alternating regularization of the diffusion process by these two regularization mechanisms. Additionally, the method constructs a set of interference operators and integrates them into the two regularization mechanisms to tackle various degradations and interferences. They provide extensive experiments to verify this work’s effectiveness across multiple fusion scenarios and under complex interference conditions.

**Questions:**

- Is this method entirely training-free, or does it still require fine-tuning? Please clarify this point.
- Why are the values set to 0 and $\hat{\bm{z}}_0^t$ respectively in Equation (13) during the alternating optimization setup?
- The paper only considers two source images as input. Is the method adaptable to more input images, such as a sequence of three images with different exposure settings for multi-exposure image fusion?
- Some minor issues include: Table 3 lacks ↑ or ↓ indicators for the metrics, and some experimental details, such as the diffusion step size, are missing.
- Besides, please address the concerns in the Weakness part.

**Ethical Concerns:**

["NO or VERY MINOR ethics concerns only"]

**Final Justification:**

I would like to thank the author for his detailed response. He explained some unclear points in the formulas and symbols, and supplemented the efficiency experiments and some additional fusion comparison experiments. These experiments and analyses resolved my concerns. I think the article is of good quality and I will maintain my score.

**Limitations:**

The authors discuss the limitations in the supplementary materials.

**Quality:**

4

**Strengths And Weaknesses:**

Strengths:
+ The idea of formulating the image fusion process as a latent diffusion process regularized by multi-source image conditions is novel.
+ Designed two regularization mechanisms, MICP and LMG, to constrain the latent diffusion process, universally applicable to various image fusion tasks without requiring training. Meanwhile, a set of interference operators is constructed, which can be integrated into the regularized mechanisms to further enhance the overall robustness.
+ The authors conduct extensive evaluations, proving the advanced performance of this work.
+ The code is provided in the supplementary material, which is commendable.
Weaknesses：
- In the Multi-source Information Consistency Projection mechanism, the authors divide the optimization into two stages: the coarse structural component $\mathcal{D}^s(\hat{\mathbf{z}}_0^t)$ and the fine detail component $\mathcal{D}^d(\hat{\mathbf {z}}_0^t)$. However, it is puzzling that the subsequent equations optimize $\mathcal{D}^s$ and $\mathcal{D}^d$ without clearly defining their relationship.
- After integrating the degradation operator $\mathbf{\Pi}$, the derivation from Equation (12) to Equation (16) is not entirely clear to me. Would it be possible to provide a more detailed explanation of this part?
- When different source images encounter different degradation interferences, how to construct $\mathbf{\Pi}$ from $\mathbf{\Pi}_i$ and $\mathbf{\Pi}_j$. Although the authors provide a brief explanation in the supplementary materials, it would be helpful if more concrete details could be provided.
- In the experimental section, the paper presents comparative results for both the base and enhanced versions of the method in infrared–visible image fusion, demonstrating promising performance; however, for multi-exposure and multi-focus image fusion, only the base version (PDFuse-B) results are reported.
- The author conducted ablation studies on the MICP and LMG mechanisms, if further provided the regularization effects of these two mechanisms at each step of the latent diffusion process would offer a more intuitive demonstration.
- Although the method achieves good results, its runtime is still relatively long (despite improvements in efficiency compared to similar diffusion-based methods).

---

> ### Author Rebuttal · Authors · 2025-07-30
>
> **Q1**: The relationship between $\mathcal{D}^s(\hat{\mathbf{z}}_0^t)$ and $\mathcal{D}^s$, as well as between $\mathcal{D}^d(\hat{\mathbf{z}}_0^t)$ and $\mathcal{D}^d$.
>
> **Reply**: In fact, $\mathcal{D}^s$ and $\mathcal{D}^d$ are shorthand notations for the coarse structural component $\mathcal{D}^s(\hat{\mathbf{z}}_0^t)$ and the fine detail component $\mathcal{D}^d(\hat{\mathbf{z}}_0^t)$, respectively. We will include this clarification in the final version for notational consistency.
>
>
> **Q2**: Explanation of the process from Eq. (12) to Eq. (16) in the LGM mechanism.
>
> **Reply**: Eq. (12) represents the base version of the LGM mechanism, which regularizes the diffusion process by enforcing consistency between $\hat{\mathbf{z}}_0^t$ and multi-source images in both the latent space and the pixel space.
> When the interference operator set is introduced, the posterior sampling is further constrained by enforcing consistency between $\mathbf{\Pi} \hat{\mathbf{z}}_0^t$ and the multi-source images. This enables the model to preserve information fidelity while also handling degradation through $\mathbf{\Pi}$, as expressed in Eq. (16). Specifically, the left-hand side of Equation (16) replaces $\hat{\mathbf{z}}_0^t$ in the pixel-space loss with $\mathbf{\Pi} \hat{\mathbf{z}}_0^t$. The right-hand side constructs an optimization objective that combines a least-squares term and a regularization term: $$\min_z \left\| \mathbf{\Pi} \mathcal{D}(\hat{\mathbf{z}}_0^t) - \max(i^m, i^n) \right\|_2^2 + \left\| \mathcal{D}(\hat{\mathbf{z}}_0^t) \right\|_2^2,$$ which is solved via the normal equation.
>
> **Q3**: How to construct the unified operator $\mathbf{\Pi}$ from $\mathbf{\Pi}_i$ and $\mathbf{\Pi}_j$ when the multi-source image inputs exhibit different degradations.
>
> **Reply**: The construction of the unified operator $\mathbf{\Pi}$ from $\mathbf{\Pi}_i$ and $\mathbf{\Pi}_j$ depends on the fusion coefficients. When the source images undergo different types of degradation, the consistency loss between $\mathbf{\Pi} \hat{\mathbf{z}}_0^t$ and the multi-source images is established by treating $\mathbf{\Pi}$ as a fused operator derived from $\mathbf{\Pi}_i$ and $\mathbf{\Pi}_j$. The fusion coefficients are determined by the spatial weights of the multi-source images during each stage of regularization.
>
> Taking the first-stage projection of the MICP mechanism in the VIS task as an example, in order to preserve the saliency of the infrared image and the majority of scene details from the visible image, the fusion is typically performed using the $\max(\cdot)$ operator. In this case, the spatial fusion weights $\text{coeff}_m$ and $\text{coeff}_n$ satisfy the relation: $$ \text{coeff}_m \cdot i^m + \text{coeff}_n \cdot i^n = \max(i^m, i^n).$$ Accordingly, the fusion of the interference operator sets $\mathbf{\Pi}_i$ and $\mathbf{\Pi}_j$ also follows this coefficient scheme. In the second stage of the MICP mechanism, the fusion coefficients for the interference operator sets are similarly derived based on the fusion weights computed over Sobel gradients. Each component of the LMG mechanism adheres to similar principles.
>
> **Q4**: For the multi‑exposure and multi‑focus image fusion tasks, only the base version, PDFuse‑B, was reported.
>
> **Reply**: PDFuse-B is the baseline version focusing solely on information fusion, while PDFuse-E is the enhanced version that integrates the interference operator set and can effectively handle various degradations in fusion scenarios.
>
> Image fusion can be divided into multi-modal image fusion and digital photography image fusion. In multi-modal image fusion, specific information metrics and loss settings are used to guide the fusion of different modality distributions, achieving the complementarity of multi-modal information. When degradations such as low light are present in multi-modal images, the information metrics in the fusion process will be affected, making it difficult to effectively aggregate the advantages of multi-modal images. Therefore, in the infrared-visible light image fusion task, we set up PDFuse-B and PDFuse-E to progressively evaluate the performance of the method, based on whether degradations are handled in multi-modal fusion.
>
> Digital photography image fusion aims to fuse multi-exposure and multi-focus images with different exposure and focus depth settings. Since exposure levels and focus depths in different source images are often complementary in different regions, PDFuse-B is enough to achieve complementary fusion of exposure levels or focus depth information in each region without the need for the enhanced version PDFuse-E with integrated degradation operators.
>
> **Q5**: A more intuitive demonstration of the regularization effects of the MICP and LMG mechanisms.
>
> **Reply**: We collected a series of intermediate diffusion results under the regularization-constrained diffusion process, $\mathcal{D}(\hat{\mathbf{z}}_0^t)$, which allows for an intuitive visualization of the effects imposed by different regularization mechanisms. From these results, it is evident that the MICP mechanism produces a pronounced enhancement of multi-source information, whereas the LMG mechanism yields an overall improvement in visual quality. Since the Rebuttal does not support posting figures, we will include this intuitive demonstration in the final version.
>
> **Q6**: Is fine‑tuning or any additional training required?
>
> **Reply**: No training or fine‑tuning is necessary. We define the multi‑source image fusion task as a regularized latent diffusion process conditioned on multi‑source images, which does not require any training or fine‑tuning of the LDMs. During inference, one needs only to inject the multi‑source images into the regularization mechanism, wherein the LDMs, guided by the regularization constraints, generate high‑quality fused images.
>
> **Q7**: The initial configuration of Equation (13) during alternating optimization.
>
> **Reply**: The MICP mechanism injects high‑fidelity information via a multi‑source image projection operator, whereas the LMG mechanism steers the distribution of intermediate latent variables through gradient guidance. Consequently, under an alternating optimization scheme, the initial settings for each mechanism differ. During MICP optimization, the LMG mechanism is disabled by setting the guiding gradient to zero. Conversely, during LMG optimization, the MICP mechanism is disabled by projecting onto the identity, i.e., setting the projection to $\hat{\mathbf{z}}_0^t$.
>
> **Q8**: Can the method support image fusion with more than two source images?
>
> **Reply**: Yes. Since our method formulates the fusion process as a regularized latent diffusion conditioned on multi‑source images, it does not impose an explicit limit on the number of inputs. Consequently, our approach can process an arbitrary number of source images in a single pass with negligible additional computational or temporal overhead.
>
> We validated this capability on the SICE multi‑exposure dataset by inputting three images with different exposure levels. It is worth emphasizing that other methods can only process two images at a time, so they perform multi-exposure fusion of three images in two separate stages via sequential pairing. In contrast, our method achieves this in a single pass. The quantitative results are presented below:
> | Method| EI↑| SF↑| EN↑| AG↑| SD↑| TE↓| SCD↑| Q_cb↑|
> |-------|----|----|----|----|----|----|-----|------|
> |U2Fusion|85.165|22.561|7.225|8.048|54.565|13.255|0.716|**0.500**|
> |DeFusion|67.214|17.257|7.279|6.206|52.841|13.201|0.550|0.357|
> |HSDS-MEF|95.637|29.464|6.994|9.367|73.821|13.486|2.989|0.351|
> |TC-MoA  |89.575|24.851|7.532|8.611|57.462|12.947|0.959|0.466|
> |CCF     |86.120|28.723|7.614|8.885|**70.408**|12.866|**3.575**|0.461|
> |SAMT-MEF|89.485|29.514|7.425|9.378|58.071|13.055|1.225|0.430|
> |EAT     |36.849|9.384 |6.501|3.146|40.089|13.937|1.700|0.347|
> | **PDFuse**| **112.422** | **30.975** | **7.708** |**10.930** |68.581(#2) |**12.771** |2.349(#3)|0.489(#2)|
>
> From the experimental results, our approach achieves leading performance on most evaluation metrics (#1 or #2), demonstrating that our regularization mechanism can be generally applied to fuse three or more source images. In addition, we further analyzed how the runtime and the Edge Intensity (EI) of the fused images produced by our method, PDFuse, change as the number of input images with different exposure settings varies. The quantitative results are presented in the table below:
> | Input sequence count |2|3|4|5|6|7|8|9|
> |-|-|-|-|-|-|-|-|-|
> | EI↑|95.155 |112.422|117.805|119.518|121.840|121.855|123.009|**124.45**|
> | Runtime (s)|18.08|18.15|18.03|18.24|18.33|18.19|18.30|18.23|
>
> It can be observed that, for the same scene, as the number of input images with different exposure settings increases, the scene information texture (EI) of the fused images gradually improves, while the runtime remains essentially unchanged.
>
> **Q9**: Time efficiency
>
> **Reply**: Compared with existing diffusion-based methods such as DDFM and CCF, our method achieves shorter sampling times while delivering superior performance. In contrast to other end-to-end fusion methods requiring multi-stage serial processing to handle more than two source images, our method theoretically requires only a single run to process an arbitrary number of source images. Therefore, these end-to-end methods will lead to a growth by multiples in runtime as the number of source image sequences grows, whereas our method maintains nearly constant runtime.
>
> **Q10**: Lack of indicator ↑, ↓ and diffusion compensation settings.
>
> **Reply**: Thanks for pointing out these issues. We will carefully address all the omissions and shortcomings in the camera ready and further improve the presentation of the figures and tables.

---

> > ### Comment · Reviewer_CeyL · 2025-08-05
> >
> > I would like to thank the author for his detailed response. He explained some unclear points in the formulas and symbols, and supplemented the efficiency experiments and some additional fusion comparison experiments. These experiments and analyses resolved my concerns. I think the article is of good quality and I will maintain my score.

---

### Note · Authors · 2025-08-12

We thank reviewers CeyL, 8Yc3, and iFAK for their feedback on concerns we have addressed; their comments have made us fully aware of the significance of the rebuttal. We also take this final opportunity to once again clarify Reviewer 2sqp’s remaining concerns regarding both the mechanism of MICP and the substantial improvements in the EI, SF, and AG metrics.

First, we would like to reiterate and emphasize the misunderstanding made by Reviewer 2sqp regarding the Laplacian operator in the MICP framework. This operator is unrelated to degradation removal or texture enhancement. In our final response to Reviewer 2sqp, we analyzed the MICP projection optimization formulation and ablation experiments to clarify that the Laplacian operator serves solely as a regularization term to balance the over-optimization of the texture layer.

Secondly, in response to Reviewer 2sqp’s concerns regarding the substantial increases in the EI, SF, and AG metrics, we provide further experimental validation targeting these indicators. Based on the LLVIP dataset and using simple Gaussian blur degradation as an example, the changes in the EI, SF, and AG metrics of the multimodal source images after adding Gaussian blur are as follows:

| Kernel Size|EI|SF|AG|
|-|-|-|-|
|Original|56.14|23.19|5.99|
|3×3|38.63|10.45|3.58|
|5×5|31.71|7.65|2.82|
|7×7|25.88|5.71|2.23|
|9×9|22.91|4.87|1.95|

As shown in the table, when the Gaussian kernel size is only 3×3, the EI, SF, and AG metrics decrease by nearly half due to the weakening of structural, frequency, and texture details, and continue to decline significantly as the degree of Gaussian blur increases. Other degradations, such as low lighting, improper exposure, and thermal diffusion, also markedly weaken the image’s structural, frequency, and texture details, thereby substantially reducing metrics such as EI, SF, and AG. When these compound degradation types cannot be fully addressed during the fusion process (for example, among the advanced methods we compared, the DDBF can only handle low-light degradation), the EI, SF, and AG metrics of the fused images also decrease significantly. Our method explicitly models various degradation types by integrating the interference operator set and solves the corresponding inverse problems using LDMs, thereby effectively addressing diverse degradation types in multimodal image fusion and generating rich details, which leads to significant improvements in metrics such as EI, SF, and AG.

---

### Decision · Program_Chairs · 2025-09-17

**Decision:**

Accept (poster)

**Comment:**

This paper proposes a training-free image fusion framework built on a pre-trained latent diffusion model, incorporating two novel regularization mechanisms (MICP and LMG). The authors provided an exceptional rebuttal that comprehensively addressed the reviewers' concerns. Consequently, most reviewers were convinced and raised their final ratings. The paper received two "Accept," one "Borderline Accept," and one "Reject." The reviewer who gave "Reject" did not raise any specific or detailed follow-up questions in response to authors' replies but instead repeatedly reiterated concerns in a general manner. After carefully reviewing all comments, the rebuttal, and the discussion, the AC recommends acceptance.